# Identification of spatially-resolved markers of malignant transformation in Intraductal Papillary Mucinous Neoplasms

Antonio Agostini[1,2], Geny Piro [1,2] ✉, Frediano Inzani[3], Giuseppe Quero[4,5], Annachiara Esposito[1,2], Alessia Caggiano[1,2], Lorenzo Priori[1,2], Alberto Larghi[6], Sergio Alfieri[4,5], Raffaella Casolino[7], Giulia Scaglione[3], Vincenzo Tondolo[8], Giovanni Cammarota[9,10], Gianluca Ianiro[9,10], Vincenzo Corbo[11], Andrew V. Biankin [7,12,13], Giampaolo Tortora [1,2] & Carmine Carbone [1,2] ✉

The existing Intraductal Papillary Mucinous Neoplasm (IPMN) risk stratification relies on clinical and histological factors, resulting in inaccuracies and leading to suboptimal treatment. This is due to the lack of appropriate molecular markers that can guide patients toward the best therapeutic options. Here, we assess and confirm subtype-specific markers for IPMN across two independent cohorts of patients using two Spatial Transcriptomics (ST) technologies. Specifically, we identify HOXB3 and ZNF117 as markers for Low-Grade Dysplasia, SPDEF and gastric neck cell markers in borderline cases, and NKX6-2 and gastric isthmus cell markers in High-Grade-Dysplasia Gastric IPMN, highlighting the role of TNFα and MYC activation in IPMN progression and the role of NKX6-2 in the specific Gastric IPMN progression. In conclusion, our work provides a step forward in understanding the gene expression landscapes of IPMN and the critical transcriptional networks related to PDAC progression.

Intraductal papillary mucinous neoplasms (IPMN) are cystic lesions of the pancreas with papillary projections characterized by a mucin-producing epithelium. The number of patients diagnosed with IPMN is increasing, with a prevalence of 0.1% per 100.000 individuals[1,2].

IPMN can originate in side branch ducts (BD-IPMN), in the main duct (MD-IPMN), or in both (mixed type IPMN) and are considered pancreatic ductal adenocarcinoma (PDAC) precursors[1,2]. MD-IPMN has a higher risk of degeneration than BD-IPMN, which in most of cases is an indolent entity[3]. In current classification IPMN are also grouped

[1]Medical Oncology, Department of Medical and Surgical Sciences, Fondazione Policlinico Universitario Agostino Gemelli IRCCS, Rome, Italy. [2]Medical Oncology, Department of Translational Medicine, Catholic University of the Sacred Heart, Rome, Italy. [3]Department of Anatomic Pathology, Fondazione Policlinico Universitario Agostino Gemelli IRCCS, Rome, Italy. [4]Pancreatic Surgery Unit, Gemelli Pancreatic Advanced Research Center (CRMPG), Fondazione Policlinico Universitario Agostino Gemelli IRCCS, Rome, Italy. [5]Digestive Surgery Unit, Department of Translational Medicine, Catholic University of the Sacred Heart, Rome, Italy. [6]Digestive Endoscopy Unit, Fondazione Policlinico A. Gemelli IRCCS and Center for Endoscopic Research, Therapeutics and Training, Catholic University, Rome, Italy. [7]Wolfson Wohl Cancer Research Centre, School of Cancer Sciences, University of Glasgow, Garscube Estate, Switchback Road, Bearsden, Glasgow G61 1BD, UK. [8]General Surgery, Department of Medical and Surgical Sciences, Fondazione Policlinico Universitario Agostino Gemelli IRCCS, Rome, Italy. [9]Department of Translational Medicine and Surgery, Università Cattolica del Sacro Cuore, Rome, Italy. [10]Department of Medical and Surgical Sciences, Gastroenterology Unit, Fondazione Policlinico Universitario Agostino Gemelli IRCCS, Roma, Italy. [11]Department of Diagnostics and Public Health, University of Verona, 37134 Verona, Italy. [12]West of Scotland Pancreatic Unit, Glasgow Royal Infirmary, Glasgow G31 2ER, UK. [13]South Western Sydney Clinical School, Faculty of Medicine, University of New South Wales, Liverpool NSW 2170, Australia. ✉e-mail: geny.piro@policlinicogemelli.it; carmine.carbone@policlinicogemelli.it

according their morphology in Gastric, Intestinal, and Pancreatobiliary, where Gastric-like cysts are usually regarded less aggressive than the others[4]. An increasing number of studies instead show that these three morphologies may represent different stages of malignant transformation rather than separate entities[4,5].

Although the morphological subtype may be indicative of the likelihood of developing a tumor, the severity of dysplasia better predicts risk of malignant transformation[6,7].

The goal of IPMN management is to reduce the risk of patient death due to progression to PDAC through primary and secondary prevention (early detection and risk-reducing surgery, respectively). High-risk IPMN (i.e. high-grade, HGD or MD IPMN, accounting for 57-90% of cases) are resected while low-risk IPMN (6-46%) undergo surveillance for the development of malignant features[8] based on morphological criteria[9]. However, the management of IPMN remains a major challenge as high or low-risk IPMN are defined based on imaging and clinical features only, not taking into account the biology underlying similar appearing lesions that ultimately drives clinical behavior. As a result, patient risk stratification is often inaccurate leading to suboptimal treatment. Around 1–11% of patients with low-risk IPMN, who were assigned to clinical follow-up, developed PDAC[10,11]. It is therefore of paramount importance to improve the understanding of IPMN biology and malignant potential to improve prognostication and personalized treatment-decision-making processes. The availability of markers predicting malignant transformation might help to stratify patients who require pancreatic surgery, which is invasive and associated with a high rate of adverse events[12].

In this study, we perform a morphomolecular analysis of a comprehensive IPMN series, comprising all disease stages and morphology of IPMN using integrated spatial transcriptomics (ST) analysis to investigate the key hallmark pathways and cell-type specific signatures associated with IPMN progression. Using two different ST technologies we map with high-resolution the whole transcriptome of different IPMN cellular types that arise during progression in their physiological context. Low-grade IPMN selected from Formalin Fixed Paraffin Embedded (FFPE) tissue from patients who never developed pancreatic cancer (follow-up > 10 yrs) were analyzed and compared to high-grade IPMN. We identify oncogenic pathways and cellular signatures able to discriminate between IPMN with different neoplastic transformation potential, distinguishing biologically low-risk from high-risk IPMN. The results of our study provide insight into the molecular characterization of IPMN and could be used to develop a molecular risk stratifier for patients with this disease.

## Results

We performed spatial transcriptomics as pictured in Supplementary Fig. S1. For Visium analysis we used an exploratory FFPE Tissue Micro Array (TMA) cohort (Fig. 1) consisting of IPMN from 14 patient including four low-grade IPMN [three low-grade-dysplasia (LGD) lesions and one Borderline IPMN], 9 high-grade IPMN characterized by high-grade-dysplasia (HGD) lesions with different histology (Gastric, $n = 5$; Intestinal, $n = 3$; Pancreatobiliary, $n = 1$). We also included four IPMN-associated PDACs and a PDAC-associated normal duct. More details about the discovery cohort samples are available in Supplementary Table S1.

We performed DNA-targeted sequencing (TSO500) on the samples from Visium exploratory cohort (Supplementary Fig. S2). All low-grade IPMN presented *RNF43* mutations, while only one sample showed *KRAS* and *TP53* mutations. All HGD IPMN presented *RNF43* and *TP53* mutations. *GNAS* mutation was found in five HGD IPMN (2 out of 5 HGD Gastric, 1 out of 3 HGD Intestinal), and in two out of four PDACs. *KRAS* was found mutated in 3 out of 5 HGD Gastric IPMN, 2 out of 3 HGD Intestinal IPMN, one HGD Pancreatobiliary IPMN and in the totality of PDAC samples. These results match the mutational profile that was already described in literature with *RNF43* mutations marking

low-grade lesions while invasiveness-associated mutations *CDKN2A* and *SMAD4* were found in HGD IPMN and PDAC[13]. No copy number variations were detected by the targeted analysis.

### Visium spatial transcriptomics data analysis and unbiased spatial clustering

After SpanceRanger pipeline processing all capture areas passed the quality control flags in the terms of quality of sequencing, gene mapping, and tissue coverage. Pancreatic tissue is known to be difficult to work with in transcriptomics because of the abundance of RNAses produced by pancreatic glands[14]. However, a mean of 61,000 reads per spot (a mean of 89 millions per capture area) were obtained from the four TMAs with valid UMI and barcodes above 98%, Q30 scores above 97%, and a mean of 15.760 genes mapped.

Before proceeding with the analysis, we conducted a verification of the alignment between pathological annotations and the IPMN. This process involved the assessment of common IPMN markers like MUC1 and CEACAM5 by two expert pathologists (as shown in Supplementary Fig. S3). However, certain challenges arose during this validation. Some tissues were inadequately covered by the spatial spots, particularly those with epithelium thickness less than $55\,\mu m$, such as the LGD IPMN in TMA1 and the ductal tissue in TMA4. Additionally, there were instances where tissues became detached or experienced degradation due to the Visium procedure (3 HGD intestinal IPMN in TMA3 and 2 HGD gastric IPMN in TMA4). These issues introduced biases that impacted the precise alignment between pathological annotations and the spatial transcriptomics (ST) clustering in TMA3 and TMA4.

We used unbiased approaches for the ST analysis rather than manual annotation of the tissue regions. We found a total of 23 spatial clusters (Fig. 2A and Supplementary Fig. S4, Source Data) in the 4 TMAs that showed correlations with histological features (Fig. 2B) using Leiden algorithm with a resolution of 0.85. This parameter was chosen after testing several resolution parameters to avoid suboptimal parameters (Supplementary Notes 1 and Supplementary Figs. S5–S8).

Five of these clusters precisely defined the different grades of IPMN, the low-grade IPMN (LGD and Borderline), and the high-grade IPMN (HGD Gastric, HGD Intestinal, and HGD Pancreatobiliary; Fig. 2B and Supplementary Fig. S4). Although LGD and Borderline IPMN shared common mutation (*RNF43*), they displayed different transcriptome. Moreover, Borderline IPMN cluster was found to be present also in few areas located in HGD IPMN. We did not identify PDAC-specific clusters due to well-recognized low cellularity of PDAC (Fig. 1) and strong admixture with stromal cells.

Seventeen clusters encompassed different types of normal and tumor-associated stroma in the 4 TMAs. We used different algorithms to infer the cell population of these clusters. Using the main molecular classification of PDAC, i.e. Moffit activated or normal stroma, we found that all IPMN shared a Moffitt-activated stroma signature regardless of their grade or morphology (Fig. 2C), indicating the presence of a typical pro-tumorigenic stroma.

### Transcriptomic dissection of IPMN identifies grade-associated transcription factors

We performed gene module analysis for the main molecular subtypes of PDAC, which correlated with Moffitt classical, Bailey pancreatic progenitor, and Collisson classical subtypes in all IPMN (Fig. 2C and Supplementary Fig. S9). Interestingly, even the low-grade IPMN showed high expression of these signatures, highlighting the presence of classical-like signatures, even in the more indolent IPMN. Moreover, basal-like, squamous and quasi-mesenchymal gene sets were not expressed in any IPMN.

Using Differential Expression Analysis (DEA) analysis, we defined gene signatures and markers that characterized each IPMN cluster (Fig. 3A, B). Interestingly, among the top markers (log2 Fold Change

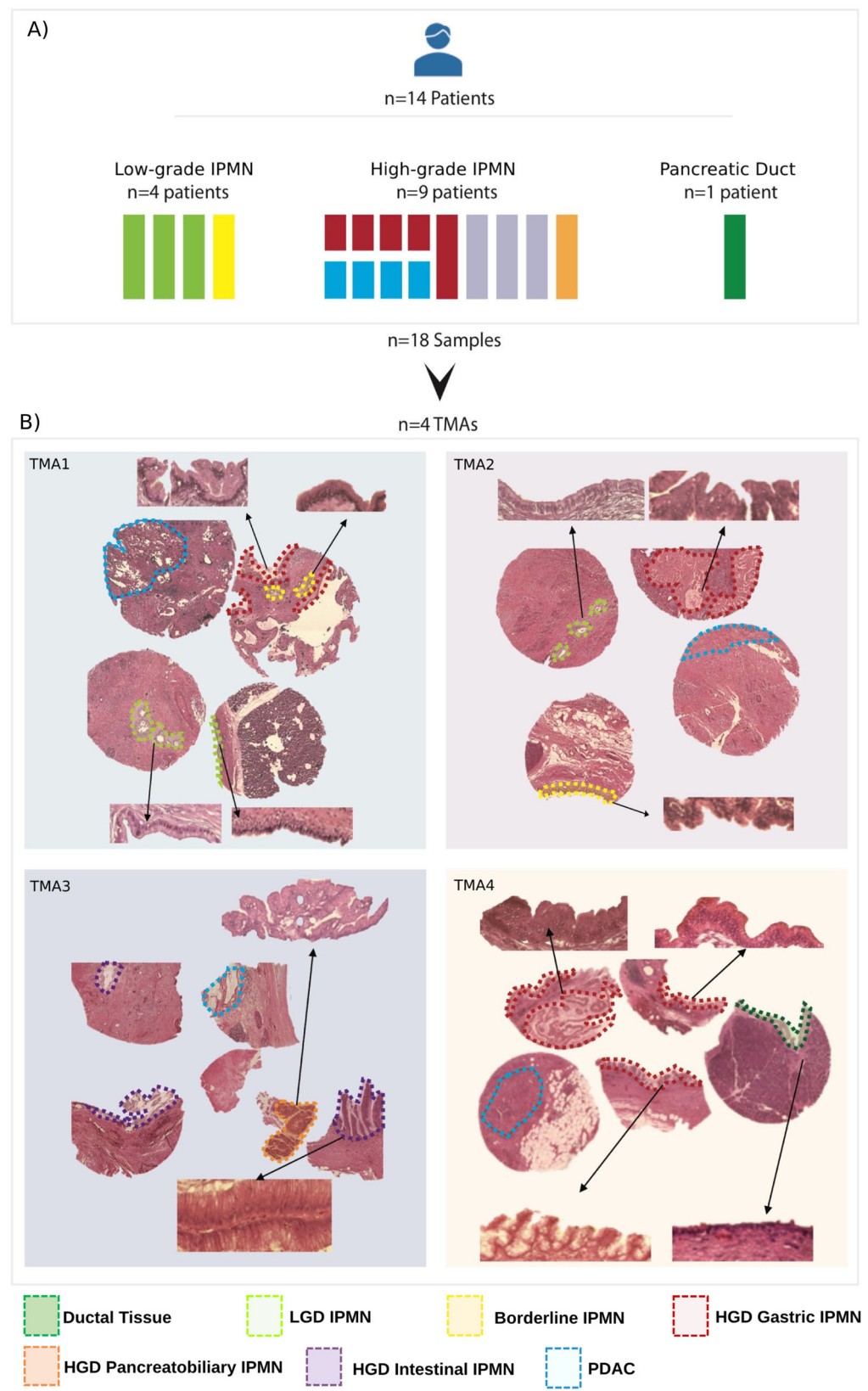

>2.5, adjusted *p*-value < 0.05, and the percentage of expression in the spots above 60%) six transcription factors associated with different clusters of IPMN (Fig. 3A). LGD IPMN was characterized by the expression of transcription factors *HOXB3* and *ZNF117*, Borderline IPMN showed high expression of *SPDEF*, *NR4A1*, and *NR4A2*; in contrast HGD Gastric IPMN was characterized by the high expression of *NKX6-2*

(Fig. 3A). Unsurprisingly HGD Pancreatobiliary IPMN shared markers with both Gastric and Intestinal IPMN. In fact several studies reported that Pancreatobiliary IPMN shares features or may originates from other morphotypes[4,15].

We also computed module scores for IPMN signatures (as shown in Fig. 3B), which were derived from these markers associated with

**Fig. 1 | Composition of discovery cohort. A** Schematic representation of the discovery cohort, which included 14 patients: 4 non-malignant IPMN, 9 HGD IPMN (including 4 patients with IPMN-associated PDAC) and 1 PDAC-associated normal duct. Non-malignant IPMN included three LGD and one Borderline (Br) IPMN. PDAC-associated HGD IPMN included Gastric (n = 5), Intestinal (n = 3), and Pancreatobiliary (n = 1) histological subtypes. In four of the five patients with gastric HGD IPMN and associated PDAC, both IPMN and PDAC lesions were used for the analysis. In total, a total of 18 samples from 14 patients were analyzed. **B** Hematoxylin and Eosin staining for the four TMAs included in the discovery cohort. 20X magnification of IPMN and Ductal Tissue are shown in the Figure inlays to show the morphology. Corresponding areas are highlighted according to the legend: Ductal Tissue (dark green), LGD IPMN (light green), Borderline IPMN (yellow), HGD Gastric IPMN (red), HGD Pancreatobiliary IPMN (orange), HGD Intestinal IPMN (violet), PDAC (blue). The picture was created with Biorender.com. TMA tissue micro array, LGD low-grade-dysplasia, HGD high-grade-dysplasia, PDAC pancreatic ductal adenocarcinoma.

different types of IPMN (LGD IPMN signature: *HOXB3, ZNF117, IGFBP3, GABRP, PDLIM3*; Borderline IPMN signature: *SPDEF, NRA4A1, NR4A2, DUSP1, PGC*; HGD Gastric IPMN signature: *NKX6-2, PSCA, SULT1C2, VSIG1*; HGD Intestinal IPMN signature: *REG4, SPINK4, CLCA1, RETNLB, ITLN1*) that specifically associated with the IPMN type.

We also performed transcription factor activity analysis with SCENIC (Fig. 3C, Source Data) and found that the transcription activity of HOXB3, SPDEF, and NKX6-2 was associated with both clustering and the RNA expression of the transcription factors.

## Hallmark cancer pathways and cell-type signatures

In order to define critical pathways involved in IPMN malignant transformation, we performed a DEA analysis of the spatial clusters previously identified by Seurat using GSEA analysis (MsigDB) with emphasis on Cancer Hallmarks and cell types.

The main pathways that exhibited activation within each IPMN cluster were used to compute ssGSEA scores for individual Visium spots.

When comparing GSEA results between high-grade Gastric (Fig. 4, Source Data) or Intestinal and low-grade IPMN (Supplementary Fig. S10, Source Data), it was evident that the key cancer-related pathways activated during IPMN progression included: TNFα signaling via NFKβ and MYC targets (Fig. 4D, E and Supplementary Fig. S10), Epithelial-to-Mesenchymal Transition (EMT) (Fig. 4F and Supplementary Fig. S10), as well as KRAS signaling (Fig. 4G).

The absence of an acquired distinct morphology of LGD IPMN was also reflected in the transcriptome. In particular, whilst HGD Gastric IPMN expressed a gene signature associated with gastric-type cells (Fig. 5), in particular with gastric isthmus cells previously identified by Busslinger et al.[16] (Fig. 5C–E, Source Data) where *NKX6-2* was one of the top markers; LGD IPMN lacked ectopic expression of a cell-type signature. The intestinal-type HGD IPMN expressed a gene signature associated with intestinal goblet markers (Supplementary Fig. S10, Source Data).

Interestingly, TNFα signaling via NFKβ, EMT, and KRAS signaling were upregulated in high-grade IPMN (Supplementary Figs. S11, and S12, Source Data) compared to Borderline IPMN and, in turn, Borderline IPMN showed increased activation of TNFα signaling compared to LGD IPMN (Supplementary Fig. S13, Source Data) indicating an association with both aggressiveness and dysplasia.

In the context of cell-type specific signatures, Borderline IPMN showed expression of gastric markers, in particular a gastric neck cell signature (Fig. 5 and Supplementary Fig. S13, Source Data).

Here, using a whole transcriptome high-resolution ST technology and an unbiased approach we identified relevant markers and pathways of IPMN with different grades and morphology. We identified the association of six specific transcription factors to IPMN clusters of different grade. TNFα signaling, Myc activation, and EMT were the key hallmarks of IPMN progression. To confirm the results obtained with an unbiased approach, we performed manual annotation of IPMN clusters filtering out all the spots that were shared between IPMN and stromal cells and the spots localized in the IPMN area that underwent tissue detachment (Supplementary Notes 2, Source Data). We performed comparable analyses (DEA with DESeq2 and GSEA with clusteRprofiler) validating the markers and gene signatures (Supplementary Figs. S14–S17) that we found to be associated with the different IPMN entities using an unbiased approach.

## GeoMx ST data analysis

The results obtained using Visium Spatial Transcriptomics were validated in an independent cohort of 57 clinically annotated IPMN samples from 40 patients included in two TMAs (TMA5 and TMA6) using GeoMx spatial transcriptomic analysis (Nanostring). These TMAs were analyzed for the whole transcriptome using a segmentation strategy with nano-dissection to eliminate immune cells (CD45 + ). A range of 200 to 900 IPMN cells was selected using the expression marker PanCK to obtain high-purity transcriptomes (Fig. 6A).

We selected 80 ROIs from 46 patients for sequencing and after stringent filtering using the GeoMxTools R package, 57 ROIs from 40 patients were retained for further analysis (6 LGD IPMN, 17 Borderline IPMN, 13 HGD Gastric IPMN, 21 HGD Intestinal IPMN).

We again observed the association between specific transcription factors and IPMN grades (Fig. 6B). *ZNF117* was expressed especially in low-grade IPMN (LGD and Borderline), in contrast, high-grade IPMN expressed low levels. Moreover, we again observed the association between *SPDEF* and *NR4A1* expression with Borderline IPMN and *NKX6-2* with HGD Gastric IPMN, highlighting the potential importance of these genes for IPMN progression to PDAC.

Moreover, we performed DEA and GSEA on GeoMx data and confirmed that TNFα signaling and Myc activation, were the main oncogenic signatures in HGD Gastric (Fig. 6C, Source Data), Borderline, and Intestinal IPMN (Supplementary Fig. S18) in contrast to LGD IPMN.

Most of the genes upregulated in the different types of IPMN using Visium spatial transcriptomics were identified through GSEA to be associated with cell-type specific signatures in the validation cohort.

We confirmed the existence of consistent signatures in our validation, which included the presence of the gastric isthmus cell signature in HGD gastric IPMN, the gastric neck cell signature in Borderline IPMN, and the duodenal goblet cell signature in HGD Intestinal IPMN (Fig. 6D, E, Source Data). Conversely, these signatures were absent in LGD IPMN (Supplementary Fig. S10, Source Data), thus reaffirming the findings from the Visium analysis.

## NKX6-2 expression correlates with progression to high-grade Gastric IPMN

Next, we used ST learn to infer the spatial evolutionary trajectory between the low-grade Borderline and HGD malignant high-risk Gastric IPMN. Although Borderline IPMN did not show any morphological characteristics of gastric differentiation, we found that the gastric neck cell markers are nevertheless already expressed. Moreover, Seurat identified several HGD Gastric IPMN spots present in the Borderline IPMN that are potential foci of transformation into the HGD Gastric subtype (Fig. 2B). Therefore, for this analysis we used IPMN with a higher presence of Borderline IPMN clusters (TMA1 and 2). We re-analyzed Visium data for the two TMAs with the python module stLearn that includes a function for trajectory inference tailored for ST analysis[17]. For these reasons we preferred this algorithm to more frequently used packages such as Monocle and scVelo that were built for Single-cell RNA sequencing.

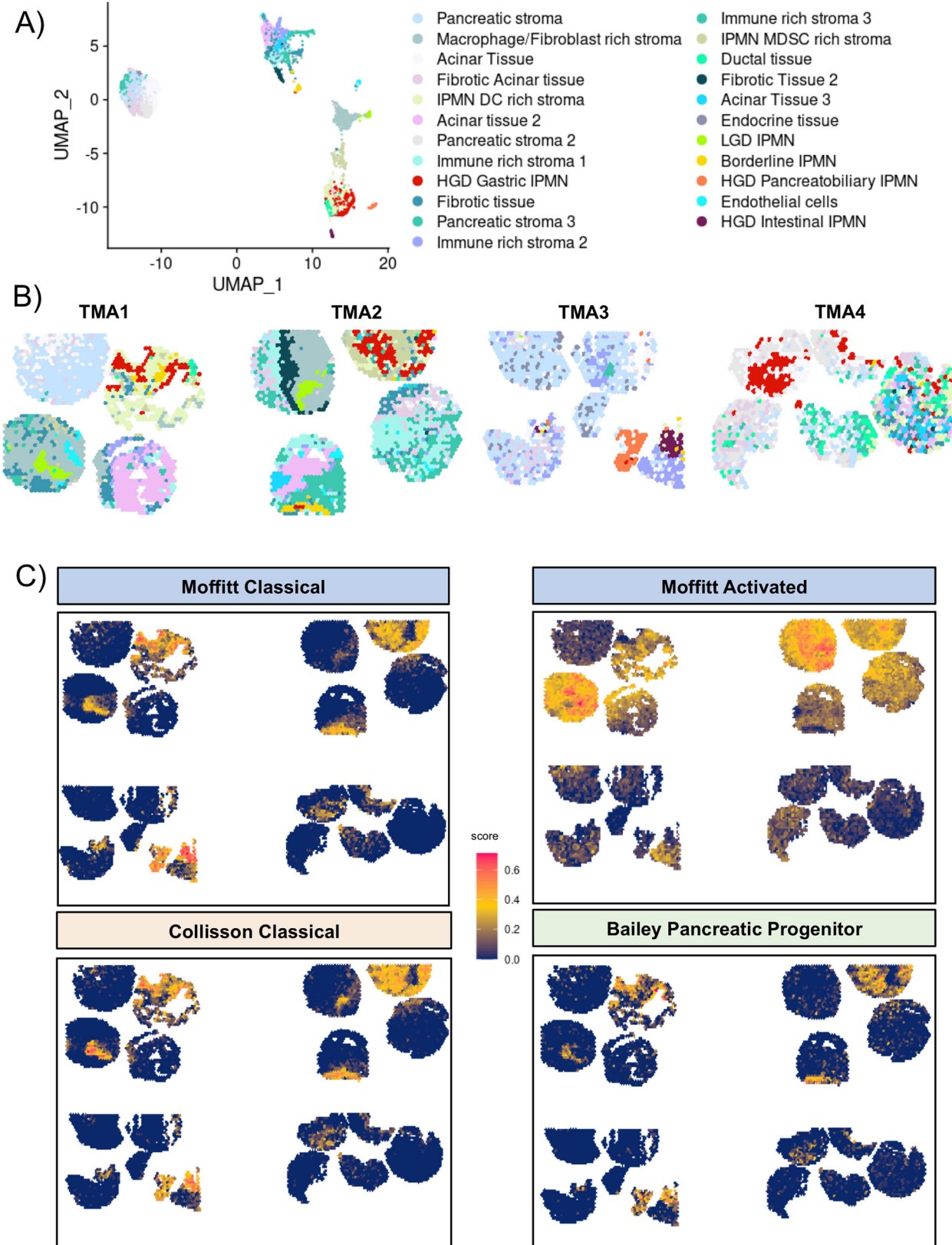

**Fig. 2 | Visium spatial features and clustering. A** UMAP plot showing the 23 clusters identified with Seurat, the clusters were annotated merging histology with ST markers. **B** Spatial visualization of the Seurat cluster alongside the 4 TMAs. Source data are provided as a Source Data file. **C** Spatial Visualization of gene module score for the main molecular signatures of PDAC: in clockwise order Moffitt Classical, Moffitt Stroma Activated, Bailey Pancreatic Progenitor, Collisson Classical. TMA tissue micro array, LGD low-grade-dysplasia, HGD high-grade-dysplasia, PDAC pancreatic ductal adenocarcinoma.

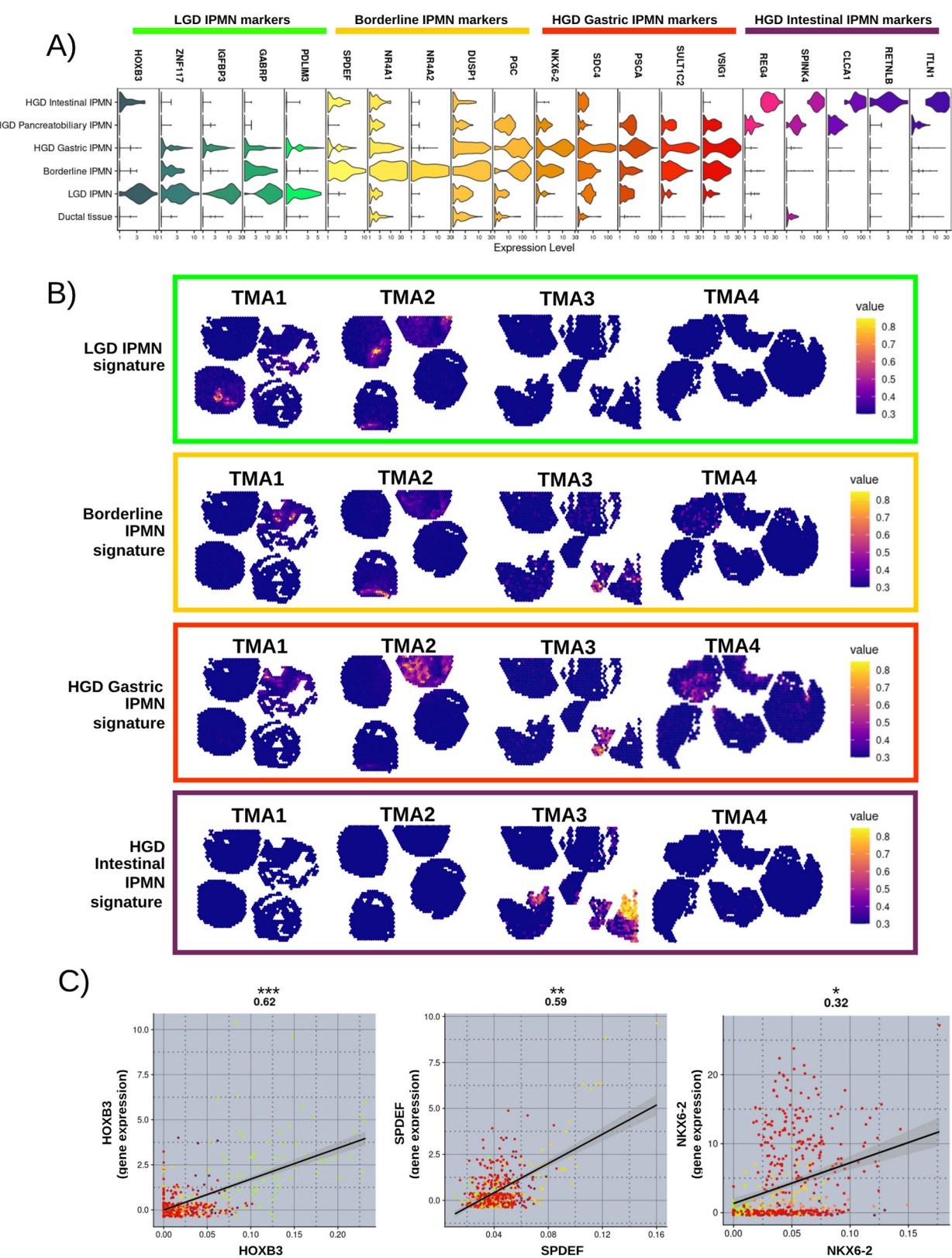

We identified 12 clusters (Fig. 7A, B), nine were stroma-related clusters and three were IPMN cell specific (1 LGD, 1 Borderline, and 1 HGD Gastric IPMN) (Fig. 7B). These three clusters matched the clusters identified with Seurat, strengthening the robustness of the results.

The spatially-aware Pseudotime analysis conducted with StLearn revealed the presence of local subclusters (Fig. 7C, D), specifically clade 69 and clade 11, within the Borderline IPMN clusters. These subclusters shared a common evolutionary trajectory leading towards HGD IPMN. We also identified the transition markers (Fig. 7E, F) for these trajectories, and notably, most of these markers were gastric-specific, aligning with our previous identification of them as specific to HGD Gastric IPMN. This included markers like *PSCA*, *VSIG1*, and

**Fig. 3 | IPMN cluster marker signatures. A** Normalized expression of the top markers of each IPMN cluster identified by DE analysis. **B** Visualization of the IPMN clusters gene markers as signatures - LGD IPMN signature: *HOXB3, ZNF117, IGFBP3, GABRP, PDLIM3*; Borderline IPMN signature: *SPDEF, NRA4A1, NR4A2, DUSP1, PGC*; HGD Gastric IPMN signature: *NKX6-2, PSCA, SULT1C2, VSIG1*; HGD Intestinal IPMN: *REG4, SPINK4, CLCA1, RETNLB, ITLN1*. Color scales indicate the score for the gene set activity of each signature. **C** Correlation between transcription factor activity (SCENIC Score) and gene expression of *HOXB3, SPDEF*, and *NKX6-2*. Two-sided Pearson correlation value was showed on the top of each correlation plot with

associated *p*-value (***<0.001; **<0.01; *<0.05). Smoothness: HOXB3 (95% CI = 0 0.56–0.67, df = 515, SE = 0.03); SPDEF (95% CI = 0.52–0.63, df = 515, SE = 0.03); NKX6-2 (95% CI = 0.23–0.39, df = 515, SE = 0.04). Each dot color refers to the annotation to the different IPMN clusters assigned (light green= LGD IPMN; yellow= Borderline IPMN; red= HGD Gastric IPMN; orange= HGD Pancreatobiliary IPMN; violet= HGD Intestinal IPMN). Source data are provided as a Source Data file. TMA tissue micro array, LGD low-grade-dysplasia, HGD high-grade-dysplasia, PDAC pancreatic ductal adenocarcinoma, CI confidence interval, df degrees of freedom, SE standard error.

particularly *NKX6-2*, which was the sole transcription factor among them. This suggests that *NKX6-2* may indeed serve as a marker for gastric-type differentiation in IPMN.

Furthermore, we conducted Diffusion Pseudotime analysis (Fig. 7G) and found a significant correlation between Pseudotime and the key transcription factors we had previously identified as markers for LGD, Borderline, and HGD Gastric IPMN. This suggests a potential transition of these transcription factors towards differentiation into HGD Gastric IPMN (Fig. 7H).

To validate the expression of the key factors identified through spatial transcriptomics in IPMN tissues, we conducted a quantitative analysis of nuclear HOXB3, SPDEF, and NKX6-2 expression in IPMN epithelial cells using the PhenoImager platform (AKOYA). This analysis was performed on an independent set of archival IPMN samples, as well as on normal pancreatic duct samples (Fig. 8, Source Data).

Our multiplex immunofluorescence (Multiplex-IF) analysis revealed that HOXB3 expression in IPMN tissues decreases as the dysplasia grade increases, and it is entirely absent in high-grade dysplasia IPMN, irrespective of the IPMN subtype (Mean: Normal Duct, 0.2%; LGD, 91.3%; BR, 43.9%; Gastric, 1.0%; Intestinal, 0.8%; PB, 0.6%).

NKX6-2 expression increases with the progression of dysplasia, with high expression observed exclusively in Gastric-type IPMN (Normal Duct, 1.6%; LGD, 1%; BR, 19.2%; Gastric, 95%; Intestinal, 1.6%; PB, 3.9%).

Conversely, while SPDEF expression also increases with dysplasia grade, it exhibits high expression exclusively in Intestinal-type IPMN (Normal Duct, 1.6%; LGD, 7.4%; BR, 30.6%; Gastric, 8%; Intestinal, 35.6%; PB, 4.4%; Fig. 8).

Based on these findings and the results of spatial transcriptomic trajectory analyses, it appears that NKX6-2 plays a crucial role as a transcriptional switch in the transformation towards a gastric phenotype in IPMN, while SPDEF is predominantly associated with the intestinal phenotype. NKX6-2 and SPDEF seem to act as exclusive and binary markers for Gastric and Intestinal IPMN, respectively. The evaluation of their expression, along with HOXB3, could offer a valuable pattern for precise clinical assessment of IPMN subtype and grading (Supplementary Table S2).

## Discussion

The identification of biomarkers that drive the progression of premalignant IPMN towards PDAC is crucial for enhancing accurate risk assessment and guiding clinical management. In this context, we propose the use of diagnostic and prognostic markers capable of distinguishing between indolent and aggressive pancreatic cysts and detecting malignant transformation. It's worth noting that routine markers for IPMN commonly used in diagnosis often fail to differentiate between non-malignant and malignant IPMN, as we also demonstrated (as shown in Supplementary Fig. S3).

Integrated whole transcriptome and high-resolution spatial profiling of the epithelial compartment of IPMN, identified and validated the main markers and pathways that distinguish cysts with different grades and morphology, and the molecular trajectory that may lead the differentiation of the Gastric type. In addition, we identified candidate oncogenic pathways that play a role in IPMN malignant

progression, particularly TNFα signaling via NFKβ, and the activation of Myc.

There are different ways to approach spatial transcriptomics, nevertheless no consensus can be found regarding the best strategies for the analyses of such data. Several algorithms were developed to integrate and analyze this type of data such as GraphST[18] and PRECAST[19]. However, for our Visium data we chose to use Harmony for data integration and Seurat for analyses, two robust methodologies that were developed for single-cell RNA-seq that proved to be powerful also in spatial transcriptomics and in fact suggested by 10X Genomics as best practice (https://www.10xgenomics.com/resources/analysis-guides/correcting-batch-effects-in-visium-data).

Here we have defined specific gene markers and signature patterns that can effectively differentiate between low-grade (LGD) and high-grade (HGD) IPMN (as depicted in Fig. 9). HOXB3 was found to be associated with LGD. HOXB3 expression plays a role in maintaining tissue homeostasis, and multiple studies have shown that its silencing can lead to a less aggressive cancer cell phenotype by restoring epithelial characteristics[20,21].

Additionally, SPDEF, NR4A2, and gastric neck cell markers were associated with borderline LGD IPMN. SPDEF is a well-established regulator of secretory cell differentiation during development[22]. In a specific study, Tonelli C. and their team used single-cell RNA sequencing (scRNA-seq) analysis to examine a mouse model of PDAC progression. They identified SPDEF as a crucial factor in an epithelial-rich cell subpopulation that is essential for the development of tumors in pancreatic epithelial and mucinous cancer cells[23]. This data reinforces our evidence as the model uses $KRAS^{G12D}$ and $P53^{R172H}$ expression throughout the pancreas (PDX1 promoter driven). Additionally, NR4A2, an oncogene, was found to play various pro-tumorigenic roles, including inhibiting apoptosis and facilitating immune escape[24].

The cell of origin of PDAC is unclear, however Pancreatic Intraepithelial Neoplasia (PanIN) studies infer progressive changes in mucin type from intestinal to gastric with PanIN of higher grade[1]. *NKX6-2* and gastric isthmus cell markers were associated with HGD gastric IPMN (high-grade). Similarly, duodenal goblet cell markers were associated with HGD Intestinal IPMN (high-grade). *NKX6-2* is a transcription factor acting on pancreas embryogenesis and specifically on endocrine progenitor cell differentiation[25]. Its expression is common in cells of gastric isthmus (Fig. 5C) and gastric subtype HGD IPMN[16], thus reinforcing the hypothesis of "paligenosis" from gastric and pancreatic ductal tissues[26].

The trajectory analysis indicated that within the context of IPMN development, various components activate multiple transcription factors, following a specific histological progression that appears to form a continuous spectrum bridging between SPDEF-high intermediate/Br and NKX6-2-high Gastric high-grade IPMN. Consequently, our results propose that NKX6-2 plays a pivotal role in driving degeneration in high-grade IPMN. This assertion is supported by recent work by Sans et al., which also highlighted the significance of NKX6-2, particularly in the gastric histotype of IPMN within the pancreas[27].

The identification of these transcriptional factors serves as a foundational step for future investigations aimed at uncovering the

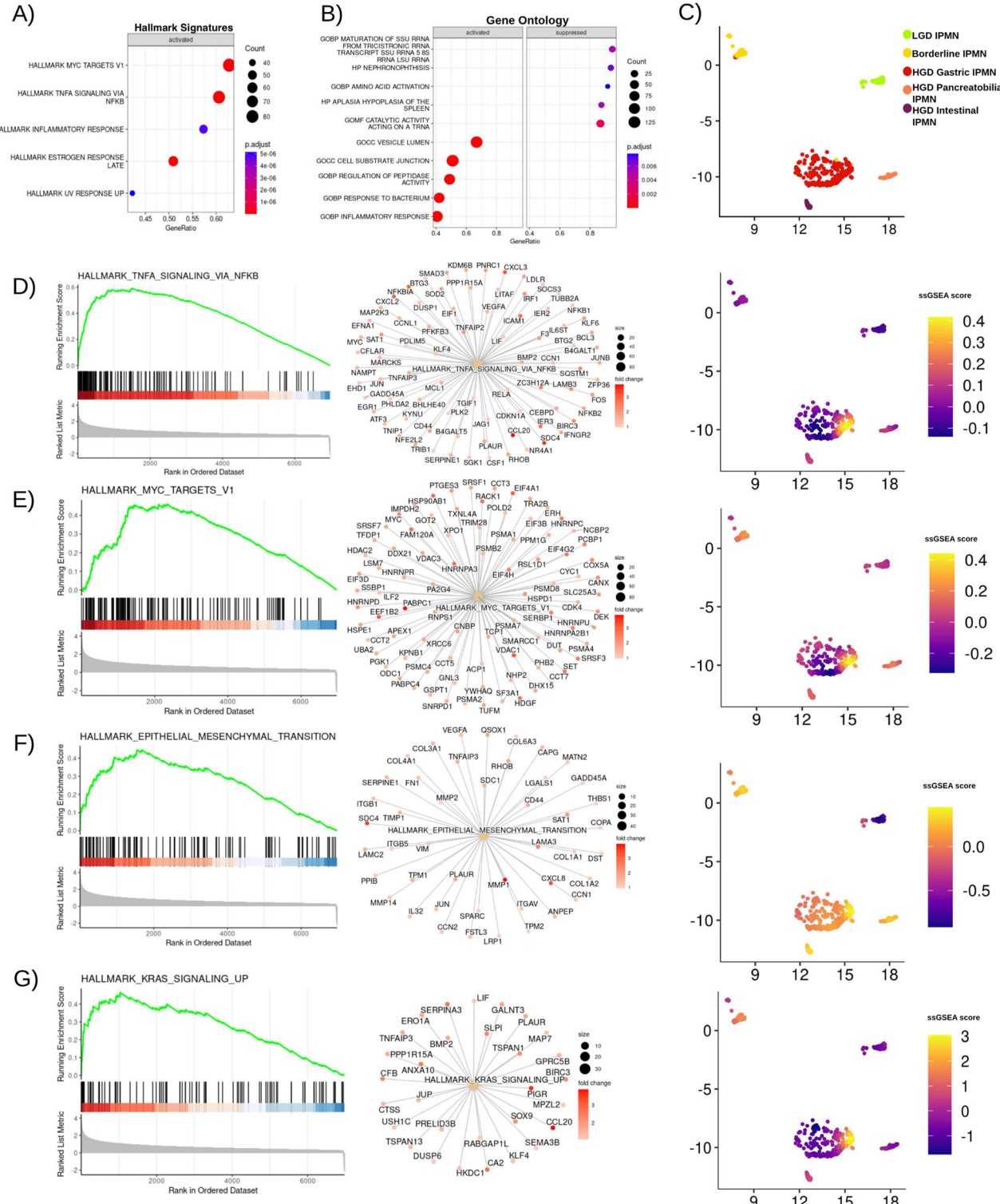

**Fig. 4 | GSEA results of comparison between HGD Gastric IPMN and LGD IPMN.**
**A** Top five Hallmark Cancer pathways activated in HGD Gastric IPMN. The circle size represents the number of genes upregulated; Two-tailed GSEA corrected for multiple comparisons with FDR < 0.05; **B** Top five activated and suppressed gene ontology signatures activated or suppressed in HGD Gastric IPMN. The circle size represents the number of genes overexpressed or downregulated. **C** UMAP plot showing IPMN clusters in 2D - dimensions. **D** GSEA plot for HALLMARK_TNFA_-SIGNALING_VIA_NFKB, network plot showing genes upregulated in HGD Gastric IPMN belonging to this signature. Featureplot showing ssGSEA score imputed for the same signature in all spots belonging to IPMN clusters. Fold changes are normalized to improve visualization. **E** GSEA plot, network plot, ssGSEA score plot for HALLMARK_MYC_TARGETS_V1. **F** GSEA plot, network plot, ssGSEA score plot for HALLMARK_EPITHELIAL_MESENCHYMAL_TRANSITION. **G** GSEA plot, network plot, ssGSEA score plot for HALLMARK_KRAS_SIGNALING_UP. Source data are provided as a Source Data file. TMA tissue micro array, LGD low-grade-dysplasia, HGD high-grade-dysplasia, FDR false discovery rate.

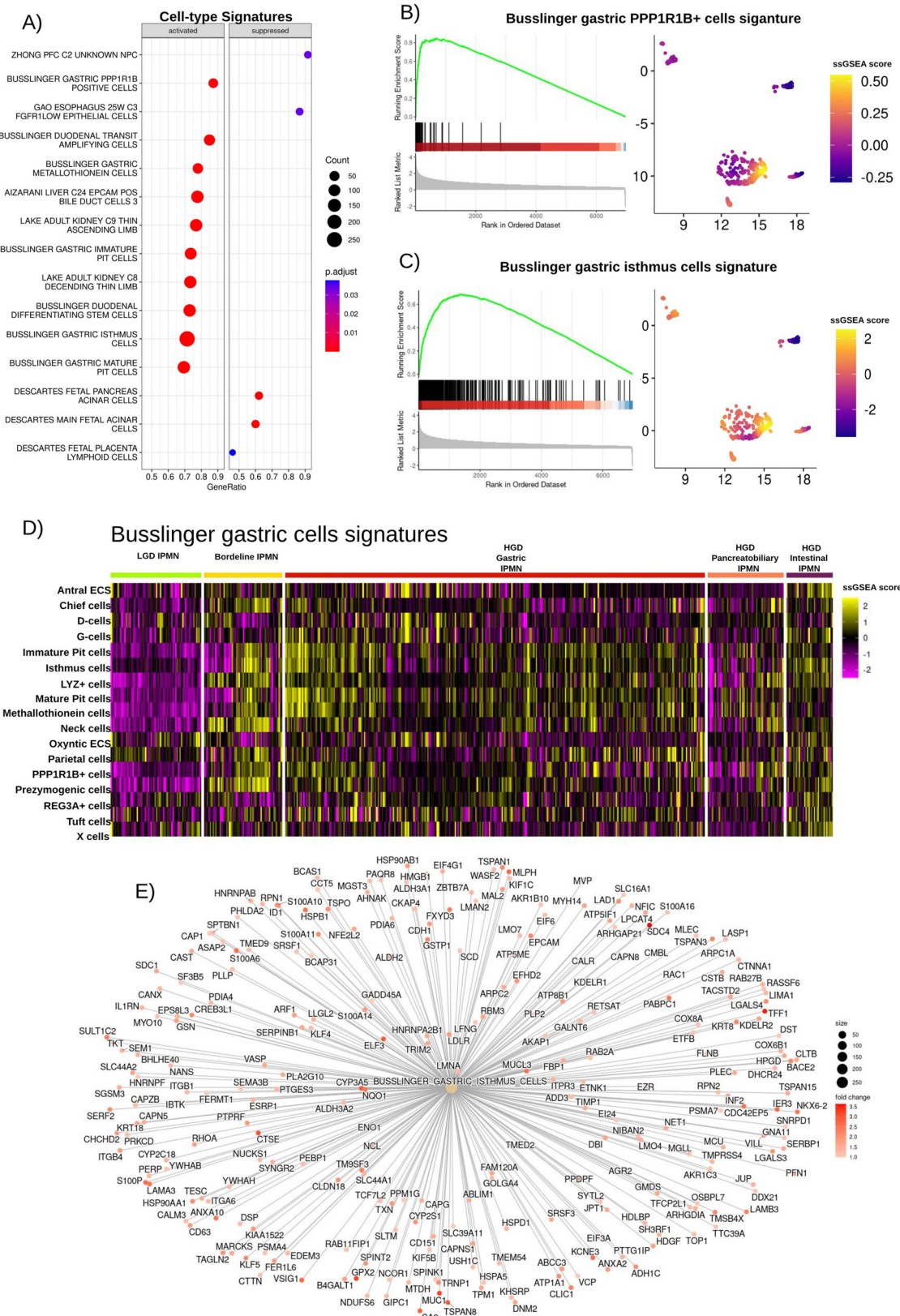

**Fig. 5 | GSEA results showing cell type specific signatures associated with HGD gastric IPMN. A** Dotplot showing the top cell type specific signatures upregulated and suppressed in HGD Gastric IPMN when compared to LGD IPMN. Two-tailed GSEA corrected for multiple comparisons with FDR < 0.05. **B** GSEA plot for the BUSSLINGER_GASTRIC_PP1R1B_POSITIVE_CELLS signature. **C** GSEA plot for the BUSSLINGER_GASTRIC_ISTHMUS_CELLS signature. **D** A heatmap displaying ssGSEA scores for each of the gastric cell signatures identified by Busslinger and colleagues, calculated for all the spots associated with IPMN clusters. Source data are provided as a Source Data file. **E** Networkplot showing the overexpression of gene included in the Busslinger Gastric Isthmus cell signature in HGD Gastric IPMN in respect to LGD IPMN. TMA tissue micro array, LGD low-grade-dysplasia, HGD high-grade-dysplasia, FDR false discovery rate.

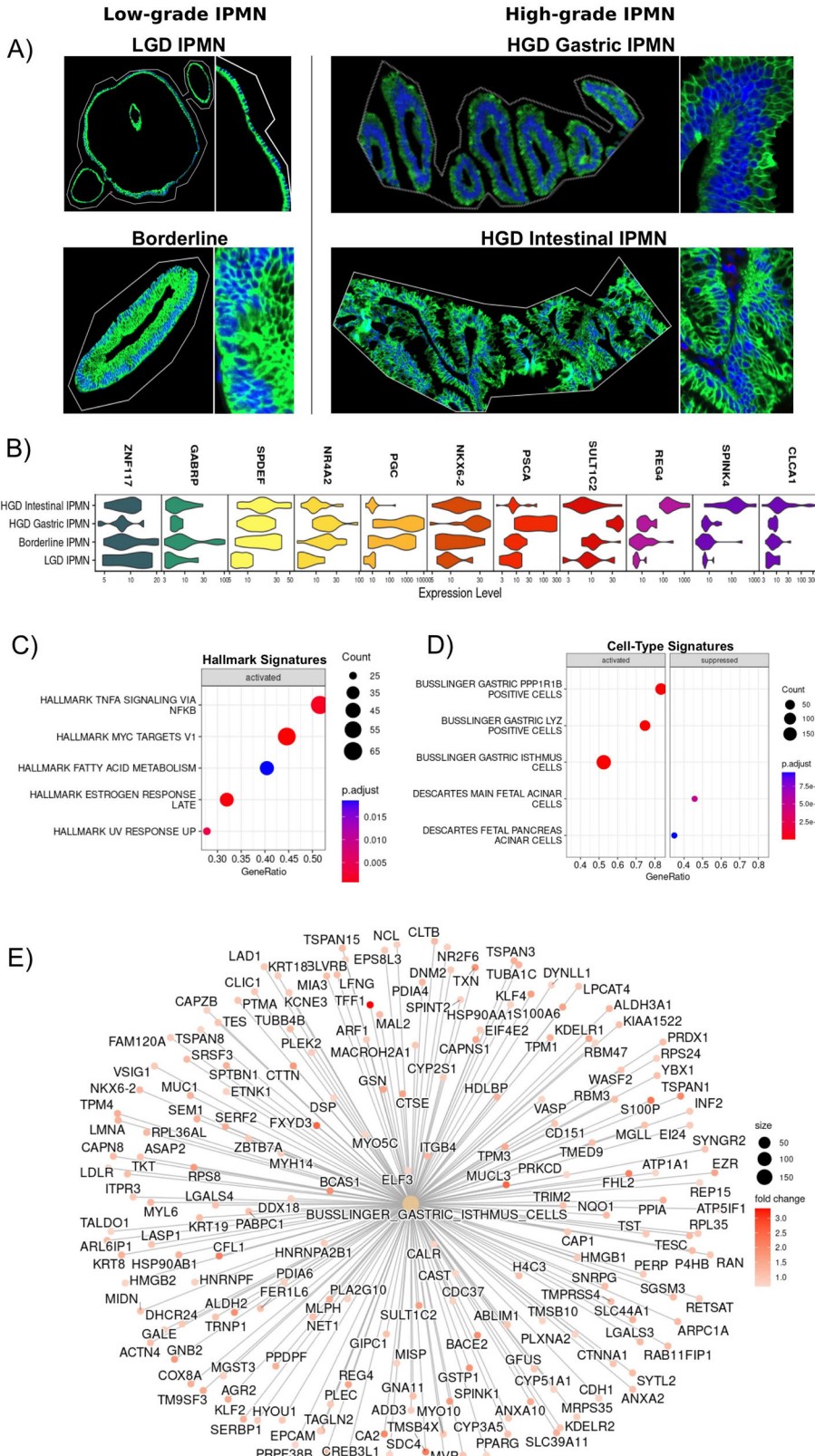

**Fig. 6 | GeoMx ST analysis. A** The image shows an ROI selected for each type of IPMN. Inlays show 20X magnification. Nuclei were stained with Syto 13 (blue), while IPMN cells were stained with PanCK (green). The picture is representative of 57 ROI: 23 low-grade IPMN (6 LGD, 17 Borderline) and 34 high-grade IPMN (13 HGD Gastric IPMN, 21 HGD Intestinal IPMN); **B** Top markers identified with Seurat are consistent with the expression markers identified with Visium. **C, D** Dotplot showing Hallmark Cancer pathways and cell type signature upregulated in HGD Gastric IPMN when compared to LGD IPMN. Two-tailed GSEA corrected for multiple comparisons with FDR < 0.05; **E** Network plot confirming the expression of gastric isthmus cell signature in HGD Gastric IPMN in respect to LGD IPMN. Source data are provided as a Source Data file. TMA tissue micro array, LGD low-grade-dysplasia, HGD high-grade-dysplasia, FDR false discovery rate.

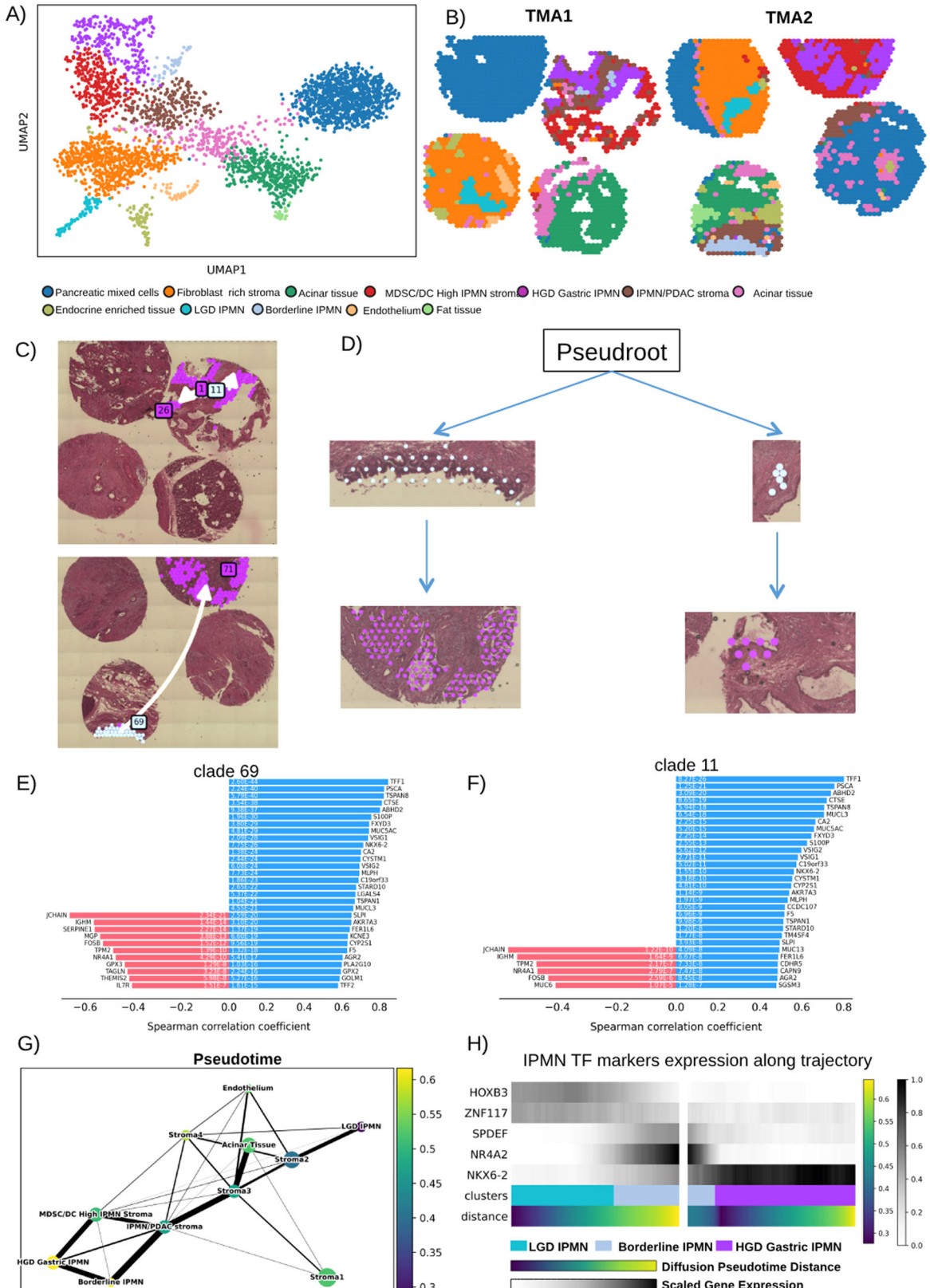

**Fig. 7 | stLearn clustering and spatial trajectory of gastric IPMN. A** UMAP plot showing clustering identified by stLearn on TMA1 and TMA2. **B** Spatial visualization of stLearn clusters. **C**, **D** show the trajectories identified leading from Borderline IPMN local sub-clusters clade 69 and clade 11 (Angel blue) to HGD Gastric IPMN (Purple). Spearman Coefficient Correlation of transition markers identified to be associated with trajectory toward HGD Gastric IPMN of clade 69 (**E**) and clade 11 (**F**).

**G** Diffusion showing the association between Pseudotime showing the evolution from LGD to HGD Gastric IPMN. **H** Heatmap showing the correlation with the expression of the transcription factors that we have identified to be markers of LGD IPMN (*HOXB3, ZNF117*), Borderline IPMN (*SPDEF, NR4A2*), HGD Gastric IPMN (*NKX6-2*). MDSC myeloid derived suppressor cell, DC dendritic cell, TMA tissue micro array, LGD low-grade-dysplasia, HGD high-grade-dysplasia.

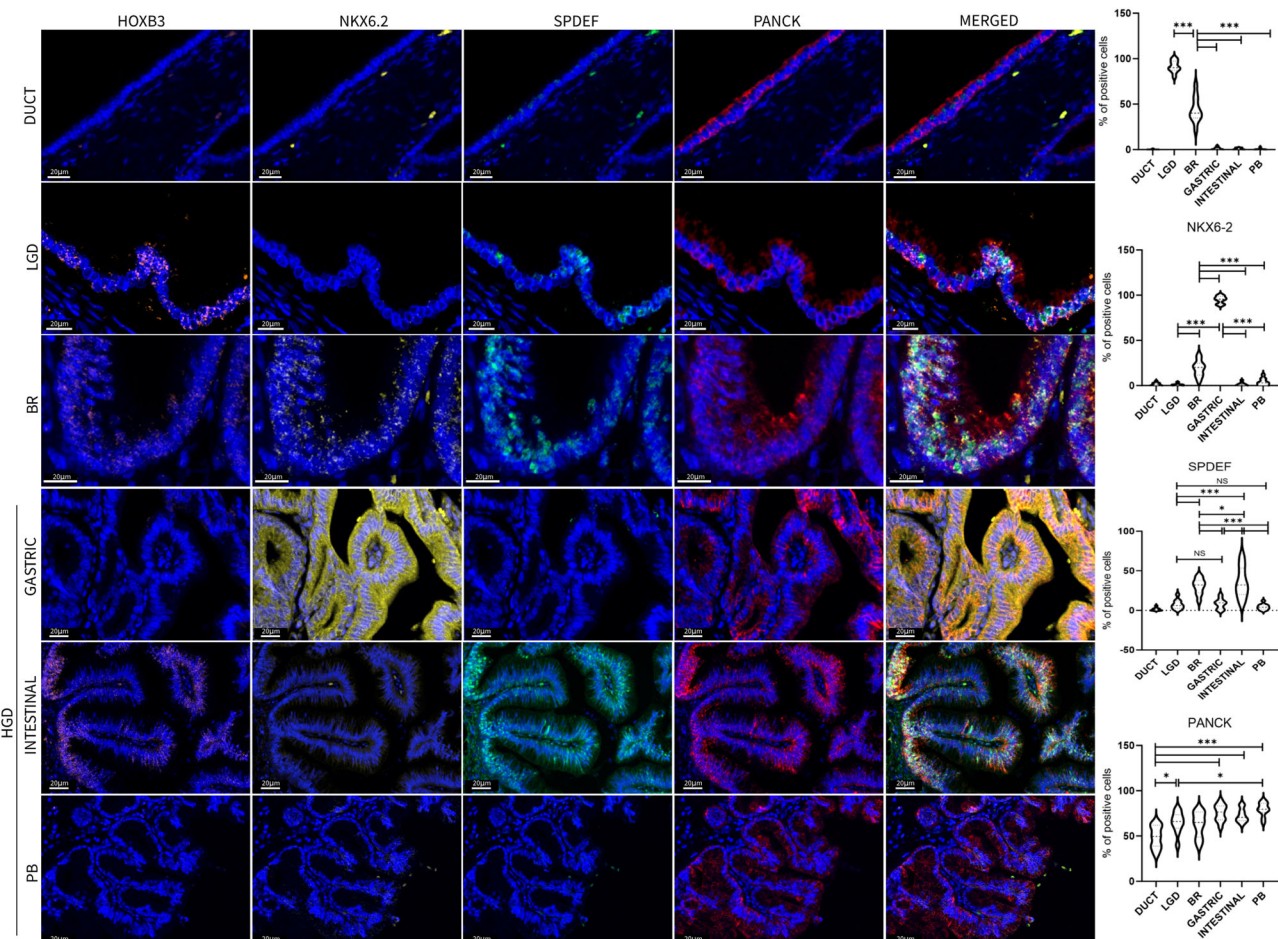

**Fig. 8 | Multiplex Immunofluorescence validation of the main markers identified by ST analysis.** Images shown are representative of 1 out of more than 10 fields acquired for each case and reviewed by pathologist. Scale bars of 20 µm are indicated in micrograph. Bar-plots show percentage of IPMN epithelial positive cells. Tukey's multi comparisons test was used to compare the differences among IPMN types (Bonferroni adjusted *p*-value ***<0.001, **<0.01, NS = not significant, df = 54). Source data are provided as a Source Data file. TMA tissue micro array, LGD low-grade-dysplasia, BR borderline, HGD high-grade-dysplasia, PB Pancreatobiliary.

mechanisms orchestrated by these factors during the progression of PDAC, originating from IPMN.

We identified oncogenic pathways that activate, during IPMN progression, TNFα signaling via NFKβ with subsequent activation of Myc. Zhao et al.[28] showed that TNFα is upregulated in PanINs, as well as in PDAC. Indeed, it is well known that TNFα promotes PDAC in many ways from supporting the desmoplastic reaction, chemoresistance[28,29], and immune escape[30]. The role of Myc in IPMN is unknown. Only recently, Kato et al. showed that HNF1β, a transcription factor commonly found in IPMN, supports IPMN proliferation by also activating Myc[31]. These data support our findings on patient samples highlighting the key role of Myc activation during IPMN progression.

This work has several strengths. It is based on the combination of technological advances of Next-Generation-Sequencing with high-resolving power imaging. This modern technique allows systematically measuring the levels of expression of all genes and to spatially resolve and associate gene expression to the specific cell, group of cells, or tissue architecture[32]. Numerous papers proposed mutational aberrations[33], extracellular vesicle proteins[34] and proteins[35], as the main transforming factors of IPMN. However, all attempts to translate the putative-identified markers of IPMN malignancy into clinical settings have been disappointing. This is likely due to the complexity of pathology, with small heterogeneous cysts and limitations of the technologies both for the number of analytes and for the type of analyzed tissues[36,37].

Although our study lacks functional experiments that validate the role of the transcription factors and oncogenic pathways in IPMN progression, we present robust data obtained from the analysis of a series of IPMN samples from a total of 69 patients, including an external validation cohort obtained from ICGC. We conducted an unbiased study on degree of dysplasia and histological features of each IPMN. In other reports, LGD and HGD areas were selected from the same IPMN sample describing in fact transcriptomic changes inside the regions with different dysplasia but from the same advanced lesion and not between two cysts at different progression[38]. To further validate HOXB3, SPDEF and NKX6-2 as markers of IPMN types, a multiplex-IF was performed on independent cohort of archival IPMN samples and on normal pancreatic ducts.

Up until now, managing IPMN requires finding the right balance between avoiding unnecessary surgeries and promptly identifying and treating patients with high-risk lesions. Despite considerable efforts, we still lack reliable biomarkers that can distinguish between IPMN cases that will progress and those that will remain inactive diseases. The markers we have identified (on both transcriptional and protein level) appear to be linked to the evolutionary trajectory that leads from low-grade lesions to high-grade IPMN. They may represent a valuable clinical tool in predicting the outcome of IPMN cases that haven't yet developed specific features, like borderline IPMN. Prospective studies will be, however, necessary to assess the efficacy of these markers.

In conclusion, in this work, we provide a step forward in understanding the gene expression landscapes of IPMN and the critical

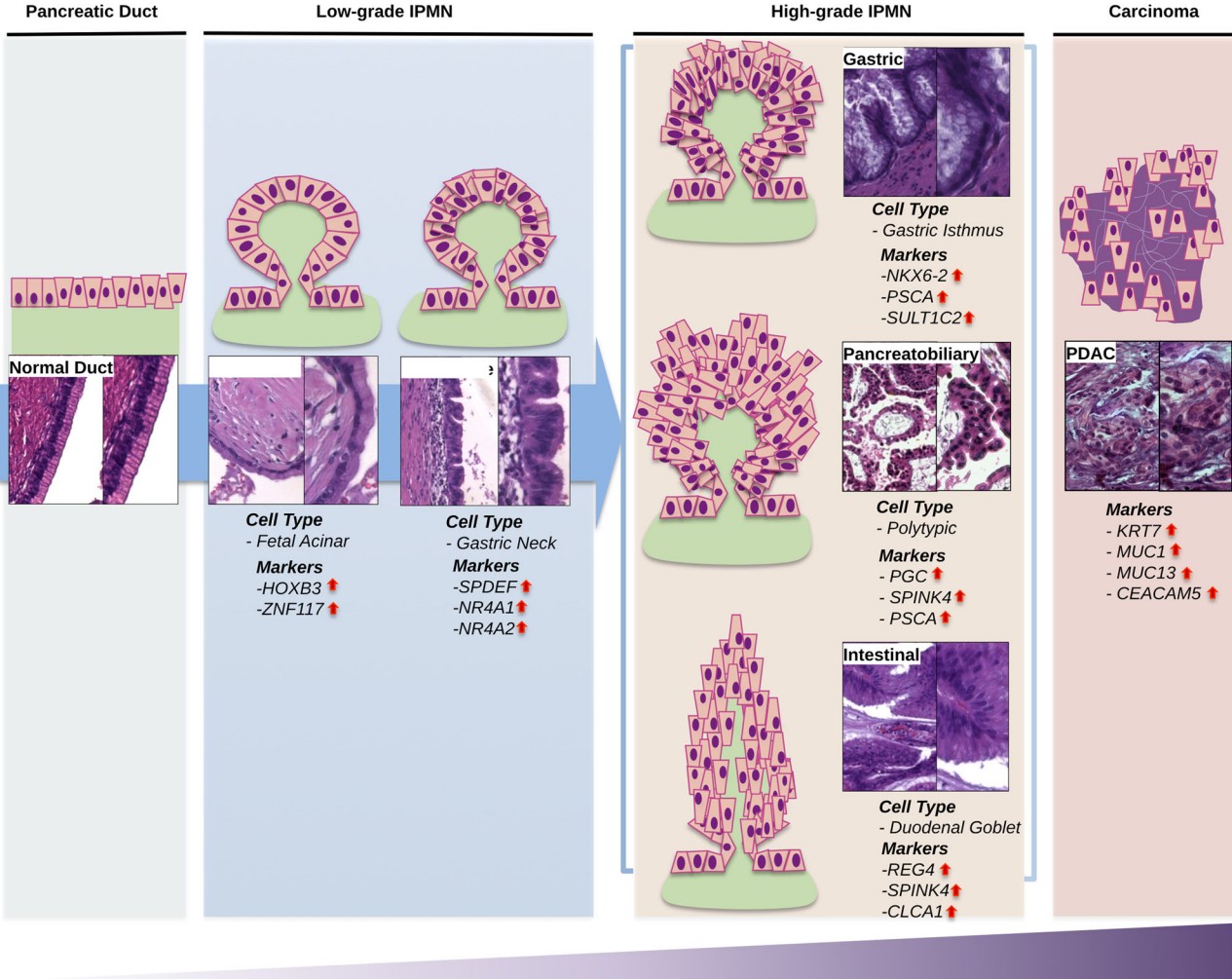

**Fig. 9 | Visual summary of the main results.** The Figure shows the main upregulated gene markers and cell-type signatures in pancreatic normal duct, Low-grade IPMN (LGD IPMN, and borderline), High-grade IPMN (Gastric, Pancreatobiliary, Intestinal), and pancreatic ductal adenocarcinoma (PDAC), together with a representative immunohistochemistry capture of each condition. Epithelial-to-mesenchymal transition (EMT), Myc, TNFα/NFKβ and inflammation signaling pathways are consistently upregulated in this malignant degeneration path.

transcriptional networks related to PDAC progression. This provides an opportunity to translate into the clinic prognostic markers for better risk stratification and management of patients with IPMN.

## Methods

### Patients and clinical sample collection

Clinical samples collection was approved by the local ethics committee (Fondazione Policlinico Gemelli IRCCS, Ethical Committee approval Prot. Gen. 3536) and followed EU regulations. All participants provided written informed consent for sample collection and subsequent analyses publication. Samples from clinically annotated patients who underwent surgery for IPMN or IPMN-associated PDAC at Fondazione Policlinico Gemelli IRCCS (Rome, Italy) from 2010 to 2021 were used for the discovery analysis (Supplementary Table S1). Archival FFPE tissues were retrieved from surgical specimens. Data collection was performed retrospectively. The independent validation was done in IPMN samples obtained from the Australian Pancreatic Cancer Genome Initiative (APGI). No information about sex or gender was collected as it is not relevant for this study.

**Discovery cohort.** The exploratory cohort consisted of 14 patients, including four with low-grade IPMN, 9 high-grade IPMN characterized by HGD (of whom 4 had IPMN-associated PDAC) and PDAC-associated normal duct (n = 1). Low-grade IPMN included three LGD lesions and one Borderline IPMN with an intermediate grade of dysplasia. High-grade IPMN included invasive lesions characterized by a high-grade of dysplasia representing the three morphotypes: Gastric (n = 5), Intestinal (n = 3) and Pancreatobiliary (n = 1). More details about the discovery cohort samples are available in Supplementary Table S1.

In 4 of the 5 patients with Gastric HGD IPMN and associated PDAC, both IPMN and PDAC lesions were analyzed. Therefore, a total of 18 samples from 14 patients were included in the analysis (Fig. 1, upper panel). All samples were sectioned, stained with Hematoxylin & Eosin (H&E) and evaluated by two expert pancreatic pathologists to identify areas containing IPMN lesions as well as dysplasia grade, morphology, progression stage, and cellularity (>30%). All these samples had good RNA quality with a DV200 score (>50%). The IPMN/PDAC areas identified by pathologists in each sample were included in Tissue Macro Array (TMA) using 1.5 mm core biopsies and the robotic TMA builder Galileo (ITS, Italy) (Fig. 1). We built a total of four TMAs: TMA1 containing two LGD IPMN, one HGD Gastric IPMN and one PDAC; TMA2 containing one LGD IPMN, one Br IPMN, one HGD Gastric IPMN and one PDAC; TMA3 with three HGD Intestinal IPMN, one HGD Pancreatobiliary IPMN, and one PDAC; finally TMA4

with three Gastric IPMN, one PDAC-associated normal duct, and one PDAC (Fig. 1, lower panel).

**Independent validation cohorts.** The GeoMx validation cohort was composed of 101 tissue cores from 61 patients. IPMN samples were obtained from the APGI, as part of the International Cancer Genome Consortium (ICGC). Two TMAs (named TMA5 and TMA6) with 101 clinically annotated IPMN cores were collected including LGD, Borderline, and HGD IPMN of different morphology. These TMAs were used for ST with the Nanostring GeoMx Digital Spatial Profiler. All TMAs were stored at −80 °C sealed with silica gel beads to preserve RNA integrity and avoid oxidation.

For markers validation, we conducted Multiplex-IF analysis using a set of archival IPMN samples, comprising 16 specimens in total. This cohort included 4 low-grade cases, 4 HGD Gastric subtype cases, 4 HGD Intestinal subtype cases, and 4 HGD Pancreatobiliary subtype cases. Additionally, 2 samples of normal ducts were included in the analysis.

The experimental protocol was approved by the local ethics committee (Fondazione Policlinico Gemelli IRCCS, Ethical Committee approval Prot. Gen. 3536) and followed EU regulations.

## Targeted genomic profiling

DNA was extracted from Visium discovery cohort IPMN samples using the Quick-DNA FFPE miniprep kit (Zymo Research) and analyzed with TruSight Oncology 500 (TSO500) (Illumina) to assess the mutational status of 500 cancer-associated genes. The Sequencing was performed on Illumina NovaSeq 6000 (Illumina) and analyzed with Trusight Oncology 500 local app v2.2 following the proprietary guidelines and parameters.

## Spatial transcriptomics

**Visium spatial.** TMAs 1 to 4 were used for Visium Spatial Transcriptomics using Visium Spatial for FFPE Gene Expression Starter Kit, Human Transcriptome (10X Genomics, USA) following the manufacturers protocols and recommended third party reagents. Visium spatial libraries were sequenced with NextSeq 550 (Illumina, USA) at a coverage of 140 million paired-end reads for each capture area according to manufacturer instructions.

An average of 89 million reads were acquired per capture area, and an average of 15,760 genes were successfully mapped, with valid unique molecular identifiers (UMI) and barcodes exceeding 98% accuracy. Additionally, the Q30 scores consistently exceeded 97%. Sequencing raw data files were obtained and processed with the Space Ranger 1.3.1 Pipeline from 10X Genomics. Space Ranger output files were imported in R with the STUtility 1.1.1[39] package and mapped to the reference H&E image. The imported outputs from each capture area were converted to a Seurat 4.3.0.1 object[40] and integrated using the Harmony 1.1.0 algorithm[41]. The integrated dataset was analyzed with Seurat and clustering was performed using the Leiden algorithm. Markers from each cluster were identified with FindMarkers() function. To infer stromal cluster composition, cell type scoring was performed on stroma clusters using the RunAzimuth() and AddModuleScore_UCell() functions from the R packages Azimuth and AUCell 1.22.0 respectively, using the Azimuth and PanglaoDB[42] gene signatures as reference. DEA between IPMN clusters was performed using Find-Markers() function whose output (only genes with adjusted p.value < 0.05) was used for Gene Set Enrichment Analysis (GSEA) with the R package clusteRprofiler 4.8.3 interrogating the MsigDB database. Single samples GSEA (ssGSEA) was performed in all spots with the R package escape 1.10.0. Transcription factor activity was assessed with pySCENIC 0.12.1[43]. To perform Spatial trajectory inference between Borderline IPMN and HGD Gastric IPMN we used the stLearn 0.4.0[17] Python library. Space Ranger outputs were imported in Python and integrated with the module for the Harmony algorithm. Subsequently,

integrated data was clustered using the Leiden algorithm using Scanpy 1.9.4 and Spatial trajectory inference analysis was performed.

Manual annotation was performed with STUtiliy ManualAnnotation() function removing the spots shared by IPMN cells and stroma; and the spots localized in the detachment region. The spots were integrated with SCTransform function while DEA between IPMN clusters was performed using FindMarkers() function. Only genes with adjusted p-value < 0.05 were used for Gene Set Enrichment Analysis (GSEA) with the R package clusteRprofiler interrogating the MsigDB database.

**GeoMx spatial.** IPMN TMA 5 and 6 obtained from APGI were analyzed for ST using GeoMx Human Whole Transcriptome Atlas (Nanostring, USA) following the provided protocol. TMAs were stained with GeoMx morphology kit to mark nuclei (Syto 13), neoplastic cells (PanCK), and immune cells (CD45). After staining all TMA cores that showed signs of detachment and degradation, or contained few IPMN cells (<100 nuclei) were discarded from the analysis. From the original set of 101 tissue cores from 61 patients, a total of 80 ROI from 46 patients were selected (multiple ROI from the same patient) for segmentation to isolate only the PanCK positive areas and exclude CD45 areas to obtain the specific transcriptome of IPMN cells (Supplementary Fig. S1).

The GeoMx library was sequenced with NovaSeq 6000 at a coverage of 541 million of reads. Sequencing data was uploaded to the Illumina BaseSpace hub and processed with DRAGEN to obtain.dcc files. The files were imported on R with GeoMxTools 3.4.0 R package and quality control (QC) was performed using default parameters to exclude outlier genes and low-quality ROI. 57 ROI from 40 patients with 23 low-grade IPMN (6 LGD, 17 Borderline) and 34 high-grade IPMN (13 HGD Gastric IPMN, 21 HGD Intestinal IPMN) passed QC and were normalized, converted to a Seurat object, and used for further analyses and data visualization.

An overview of the ST workflow is provided in Supplementary Fig. S1, and a more detailed methods are available in Supplementary Methods.

## Multiplex immunofluorescence

We performed multiplex IF analysis by the Opal 6-Plex Detection Kit (NEL821001KT, Akoya Biosciences) following standard protocol on a series of IPMN samples (n = 16; low-grade, 4; HGD Gastric, 4; HGD Intestinal, 4; HGD Pancreatobiliary, 4) and normal ducts (n = 2). Two expert pathologists in blind confirmed the IPMN histopathological features.

The following antibodies were used for IF analyses: HOXB3 (PA5-103890, Thermo Fisher Scientific), SPDEF (ab220776, ABCAM), NKX6-2 (ABN-1455, MERCK), and PanCK (67306S, CellSignaling).

Before proceeding, optimal staining conditions for each marker were determined using monoplex stained slides from a positive control for each antibody. Protein Atlas (www.proteinatlas.org) was used to identify the positive control tissues that express high levels of each protein. We tested the antibody for HOXB3 (PA5-103890) for IF on both IPMN and testis samples. We found the optimal dilution to be 1:100. The antibody for SPDEF (ab220776) was tested for IF on both IPMN and salivary gland samples finding an optimal dilution of 1:400. The antibody for NKX6-2 (ABN-1455) was tested on both IPMN and spinal cord samples (dilution 1:1000) and PanCK (67306 S) on both IPMN and on lung and colon cancer samples (dilution 1:1000).

Multiplex slide images were acquired by Phenoimager Workstation (Akoya Biosciences, US) and processed with QuPath 0.4.2 for cell segmentation and positive cell count; the operator performing the analysis was blinded to the diagnosis.

## Reporting summary

Further information on research design is available in the Nature Portfolio Reporting Summary linked to this article.

## Data availability

The Spatial Transcriptomics data generated in this study have been deposited in the GEO database under accession codes: GSE229877 (Visium raw and processed data) and GSE229752 (GeoMx raw and processed data) TruSight500 raw data is available on Sequenced Read Archive (SRA) BioProject SRA number: PRJNA1013719. Source data are provided with this paper.

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

## Acknowledgements

This work was supported by My First AIRC Grant (MFAG) "Luigi Bonatti e Anna Maria Bonatti Rocca" grant number 23681 to C.C.; AIRC IG 18178 and PRIN 2022 PNRR Prot. P2022LN3KS to V.C.; AIRC MFAG n.29224 to G.P.; AIRC IG 24519 to A.L; AIRC IG 26330, Ministry of Health (CO 2019-12369662), FIMP (J38D19000690001), Italian Ministry for Universities and Research (MUR) PRIN 2022 Prot. 2022P79F9N to G.T.

## Author contributions

Each author significantly contributed to the conceptualization of the study, the acquisition, analysis, or interpretation of data, as well as the drafting of the paper. All authors approved the final version of the manuscript. A.A., C.C., G.T., and G.P. designed the research; A.A., G.P., A.C., L.P., and A.E. performed the experiments; A.A., C.C., G.P., and R.C. analyzed the data and generated displays; G.Q., A.L., S.A., and V.T. collected clinical samples: F.I., G.S., and V.C. evaluated patient material; G.C. and G.I. collected clinical data, A.V.B., V.C., and G.I. gave conceptual advice; A.A., C.C., V.C., and G.P. wrote the manuscript; C.C. and G.T. supervised the study; and all authors discussed the results and implications of the study.

## Competing interests

The authors declare no competing interests.
