## [Peer Review File · Nature Communications]

EDITORIAL NOTE: This file contains images taken from the single-cell transcriptomics data of the Human Protein Atlas, version 23.0 (released on 19/06/2023). Such images have been redacted. The redactions have taken place on page 24 and 25 of the Peer Review File. The general citation for this source is the following:

Karlsson, M. et al. A single-cell type transcriptomics map of human tissues. *Sci. Adv.* 7, eabh2169 (2021).

REVIEWER COMMENTS

Reviewer #1 (Remarks to the Author): Expert in IPMN pathology and genomics

The authors investigated spatial transcriptomics (ST) in IPMNs of various subtypes focusing on their progression from low-grade dysplasia (LGD), high-grade dysplasia (HGD), and invasion. They found novel subtype specific markers: HOXB3 and ZNF11 in LGD, SPDEF in borderline, and NKX6-2 in HGD. They emphasized a role in IPMN malignant progression of THFalpha signaling via NFkB and Myc activation. This study is novel and seems to be highly significant because it provided detailed information of ST in IPMN with finding of several potential key molecules likely to play a role in malignancy progression of the neoplasm. Following concerns should be addressed.

1. It is very curious that whether the key molecules could be validated in protein level.

Immunohistochemistry could give information of the actual protein expression with evaluation of spatial associations.

2. Genotypes could be associated with transcriptome data. Associations between spatial transcriptomics and genotypes, e.g., mutations in KRAS, GNAS, TP53, and RNF43, and amplification of MYC, could be analyzed.

Reviewer #2 (Remarks to the Author): Expert in spatial transcriptomics and computational cancer genomics

The study is designed in understanding the gene expression profiles of IPMN and the essential transcriptional networks associated with the progression to PDAC. The authors used two technologies to analyze two independent cohorts of IPMN samples and identified a few markers associated with different grades of dysplasia. The identification and validation of critical signaling pathways and transcription factors that may be responsible for the malignant transformation of cystic pancreatic lesions have the potential for developing clinical prognostic markers for the improved risk stratification and personalized management of IPMN patients. The motivation of this manuscript is clear, but the corresponding ST data was not provided and the simple biological analysis in the Results section is inefficient to demonstrate the marker discovery. However,

1. The study is limited by the lack of functional experiments that would validate the roles of the identified transcription factors and oncogenic pathways in the progression of IPMN. The authors should consider addressing this limitation in future studies.

2. The authors should provide the ST data and add more convincing biological analysis.
3. Furthermore, the authors should provide more discussion on the potential clinical implications of their findings. Specifically, they should address how these markers could be used in practice to improve patient outcomes and further validation studies are needed.
4. Although there are new findings in the work, additional experimental validation is necessary to support the findings.
5. Except for very rough spatial trajectories, relevant biological validations have been performed using single-cell or bulk analysis processes, and no cases have required SRT data to complete validation. In fact, from the perspective of the marker genes discovery, single cell or bulk data can also complete the work in this manuscript, so why use SRT data.
6. Marker genes discovery requires comparison of SRT expression from different grade lesions, which involves comparability of expression data due to the batch effect of SRT data. The author should explain in detail how to eliminate the batch of SRT data to make gene expression comparable. Note that, Harmony method adopted in this paper can only compress SRT data from different sources for cluster analysis, which cannot guarantee the comparability of each gene expression data. Therefore, it is not convincing by the way of identifying marker genes in this paper.
7. GSVA analysis is rough, and relevant results need to be further validated in space, and further explained in terms of biological processes or cell type functions. In addition, it is necessary to discuss whether these finding can be observed in most of the sample data. The marker selected in this paper is mainly TF, and the author should provide discussion on relevant TF regulation network or mechanism.
8. Spatial trajectory inference can also be affected by data batch. In addition, the analysis of spatial trajectory inference is too rough, and relevant details need to be further provided. For example, pseudo-time and their display in situ, the related genes on the pseudo-time heat map display and so on.
9. The authors are unable to provide experimental verification of marker genes, whether to provide predictive analysis of these marker genes in clinical diagnosis. For example, the classification of validation ST data or other types of data.

Minor issues:

1. Does TMA refer to sections made up of tissue samples from different patients? Whether batch effect exists in one TMA and whether batch effect exists in different TMA?
2. In Fig 3B, what are the IPMN Signature?
3. In Fig 7E, what are the clade_69 and clade_11?
4. The legend in the text is too brief, the author should enrich the related legend.
5. There are some formatting or spelling mistakes in the manuscript, which should be checked carefully.

Reviewer #3 (Remarks to the Author): Expert in spatial transcriptomics, cancer genomics, and tumour microenvironment

In this manuscript, to identify novel biomarkers that could better stratify IPMN risk, Agostini et al. performed Visium spatial transcriptomics analysis on 18 core biopsies (4 TMAs) from 14 patients in the discovery cohort. They identified differentially expressed genes and pathways in LGD and HGD IPMN and further carried out GeoMx spatial transcriptomics analysis of 57 IMPN samples from an independent cohort to validate these findings. While this study may present a valuable resource for the community, this study is overall descriptive and the analysis of Visium data lacks depth, the significant analytical and conceptual limitations diminish its value and my enthusiasm for it.

Major comments:

1. In the current version of the Methods section, there's no detailed description of how the batch effects were assessed, how significant they were, and the method(s) applied for batch correction. There's also no evaluation of the performance of their batch correction method. It is known that Harmony is a method that normalizes embedding, but not data matrices.
2. How did the unbiased approaches for the ST analysis correlate with the annotation of tissue regions by pathologists? How were correlations with histological features assessed (quantified)? By eyeballing Fig. 2B and Fig. 1a, it appears TMA3 and TMA4 showed a lower correlation with pathology annotation. Fig. 2B is for visualization purposes and had to eyeball the correlation. Visualization alone isn't sufficient; I would suggest the authors show spot-level correlation.
3. The study identified 23 spatial clusters. On what basis was the resolution (the optimal number) of their clustering analysis determined? How can they ensure that the tissues weren't overclustered or underclustered?
4. In Fig. 2A, it's challenging to read the colors. It seems like the HGD intestinal IPMN formed a separate cluster, and HGD IPMN intermixed with ductal tissue.
5. Using the main molecular classification of PDAC (i.e., Moffit activated or normal stroma), how does the Moffit activated signature inform the cell population of these clusters? The Moffit activated signature does not seem to be robust in distinguishing HGD and LGD; for multiple cores, the signature was all over the place. Does the Moffit classical correlate with the PDAC classical lineage state? How about the basal state? Visualization isn't sufficient; please show spot-level correlation.
6. What are the differences between the Moffit classical, Collisson classical, and Bailey pancreatic progenitor signatures? How many genes are overlapped among these signatures?
7. On page 8, the authors stated, "Interestingly, even the low-grade IPMN showed high expression of

these signatures, highlighting the presence of classical-like signatures, even in the more indolent IPMN.” I question this as, for all three cases with low-grade IPMN, these signatures weren't only expressed by low-grade IPMN but also by other tissues. Could there be another possibility? For example, some genes of these signatures are not specific and could also be expressed by immune/stromal cells. Have the authors ruled out such a possibility? Additional validation data would be needed to support this statement.

8. In Fig. 3A, what criteria were used to select these genes? How many transcription factors were these six genes selected from, and how were they identified? Why do the authors find them “interesting”? Please provide more details.

9. Both SPDEF and NR4A1 were high in Borderline IPMN, but their expression decreased in HGD IPMN samples. How can their negative correlation with survival be explained?

10. HGD Gastric IPMN markers PSCA, SDC4, and VISIG1 showed significant variation across the histology subtypes and had the highest expression levels in gastric IPMN. Will this limit their prognostic significance? I suggest stratifying the survival analysis in Fig. 3C by histological features. Will these genes still be significant for survival after adjusting for histology?

11. How were these signatures in Fig. 3B built, and how do they differ from those in Fig. 2C?

12. In Fig. 4-5 pathway analysis, can any of these pathways be validated?

13. The analysis of Visium data lacks depth. The immune and stromal cell compartments of IPMN are largely overlooked. I would suggest running inferCNV and mapping KRAS mutations using Visium data and, if possible, profiling the immune infiltrates and stromal cell activities.

14. Why were 23 out of 57 ROIs excluded from subsequent analysis? Even if they aren't covering IPMN, they could be close to IPMN and those ROIs. Could they still be used to study the immune and stromal cells of IPMN?

15. Regarding the statement: “HOXB3 and ZNF117 were associated with LGD”. In Fig. 6B, HOXB3 was not included. ZNF117 was included but was also highly expressed in HGD. I would suggest adding normal tissue as a control, validating HOXB3, and conducting further experiments to support the “association”.

16. Functional validation is needed to support statements such as, “This infers that ZNF117 may be a key transcription factor in the early stages of IPMN,” and, “suggesting that NKX6-2 may be a major trigger of gastric-type differentiation in IPMN.” The same goes for the statements about the pathway analysis. There are many overstatements in the manuscript.

17. In Fig. 7A, how do these 12 clusters correlate with the clusters in Fig. 2? Why were these stromal cell-enriched clusters distinctly clustered? Could this be biologically relevant or influenced by batch effects? Can the authors interpret the results?"

Minor comments:

1. What was the median number of genes per spot? This information isn't provided in the manuscript or Methods. Also, what percentage of spots passed QC per core?
2. For GeoMx, in addition to panCK and CD45, any additional (the 3rd) morphology marker was used?
3. "Five of these clusters precisely defined the different grades of IPMN: the low-grade IPMNN (LGD and Borderline), and the high-grade IPMNS (HGD Gastric, HGD Intestinal, and HGD Pancreatobiliary) (Fig. 2B)." Which five clusters were these?
4. Is panCK able to label all IPMN cells? How sensitive and specific is it for GeoMx analysis?
5. In Fig. 6A, please add a color key. In Fig. 6B, please add p-values.
6. The authors said, "Moreover, we again observed the association between SPDEF and NR4A1 expression with Borderline IPMN." I didn't find the gene "NR4A1" in Fig. 6B.

Point-by-point response_NCOMMS-23-14460-T

We would like to thank the reviewers for their informative critique. We are confident that we have addressed the issues raised and feel that this has further improved the quality of the manuscript.

A copy of the manuscript, where the changes are highlighted in yellow and ~~as grey-crossed-out (eliminated text)~~ has been uploaded as Additional File for Review but NOT for Publication.

Reviewers will find below the point-by-point response, which include several figures which have been produced exclusively for the purpose of the review and therefore not included in the revised version of the manuscript.

For the new data that are instead included in the revised version of the manuscript, please see the reference to the figure number in the response to reviewers' comments.

Reviewer #1 (Remarks to the Author): Expert in IPMN pathology and genomics

The authors investigated spatial transcriptomics (ST) in IPMNs of various subtypes focusing on their progression from low-grade dysplasia (LGD), high-grade dysplasia (HGD), and invasion. They found novel subtype specific markers: HOXB3 and ZNF11 in LGD, SPDEF in borderline, and NKX6-2 in HGD. They emphasized a role in IPMN malignant progression of THFalpha signaling via NFkB and Myc activation. This study is novel and seems to be highly significant because it provided detailed information of ST in IPMN with finding of several potential key molecules likely to play a role in malignancy progression of the neoplasm. Following concerns should be addressed.

1) It is very curious that whether the key molecules could be validated in protein level. Immunohistochemistry could give information of the actual protein expression with evaluation of spatial associations.

R. We fully agree with your suggestion. The protein level of the key molecules were tested and validated by Opal Multiplex Immunofluorescence on IPMN Tissue Sections. A new picture, confirming the ST data, was added in panel of figure 8.

2) Genotypes could be associated with transcriptome data. Associations between spatial transcriptomics and genotypes, e.g., mutations in KRAS, GNAS, TP53, and RNF43, and amplification of MYC, could be analyzed.

R. This is a great point. We analyzed more than 500 hotspots for somatic mutations by targeted sequencing (Trusight Oncology 500, TSO500) on IPMN

samples. Please, find the genomic data in Supplementary Figure 2. Specifically, since the quality of DNA extracted from the FFPE samples was not enough to perform WES in all samples, we used TSO500 panel that represents clinical routine in our institution for FFPE sample genomic screening. Moreover, the panel comprises the relevant IPMN and PDAC genes including *KRAS*, *GNAS*, *TP53*, *CDKN2A*, *SMAD4* and *RNF43*. As expected, low-grade IPMN had *RNF43* mutations which are typically associated with this early lesion; high-grade IPMN and PDAC shared instead more advanced and oncogenic mutations such as *KRAS*, *TP53*, *CDKN2A*, and *SMAD4*. *MYC* amplification was not detected in all of the analyzed samples and this could be due to the known limitations of targeted sequencing analysis. We added more details in the results section.

Thank you again for this on point comment, we firmly believe that helped us to improve our results.

Reviewer #2 (Remarks to the Author): Expert in spatial transcriptomics and computational cancer genomics

The study is designed in understanding the gene expression profiles of IPMN and the essential transcriptional networks associated with the progression to PDAC. The authors used two technologies to analyze two independent cohorts of IPMN samples and identified a few markers associated with different grades of dysplasia. The identification and validation of critical signaling pathways and transcription factors that may be responsible for the malignant transformation of cystic pancreatic lesions have the potential for developing clinical prognostic markers for the improved risk stratification and personalized management of IPMN patients. The motivation of this manuscript is clear, but the corresponding ST data was not provided and the simple biological analysis in the Results section is inefficient to demonstrate the marker discovery. However,

1. The study is limited by the lack of functional experiments that would validate the roles of the identified transcription factors and oncogenic pathways in the progression of IPMN. The authors should consider addressing this limitation in future studies.

R. We recognize that a part of functional validation is missing. Further studies are planned to establish the specific role of IPMN markers, associated with different histology, in tumor evolution and progression. In particular, our plan is to validate the molecules identified in this study in a relevant model of IPMN. For this reason we are currently developing an array of organoid cultures derived from human IPMNs associated with pancreatic cancer (malignant IPMNs). We planned to knock-out the candidates herein identified and assess the consequences of the perturbation on the malignant behaviour of these cultures. Noteworthy, while our manuscript was under review, Sans and colleagues validated the transcription factor *NKX6-2*, which we link here to the

gastric histotype, as a key factor in maintaining the gastric IPMN in the pancreas. This is an important point that further confirms our data.

2. The authors should provide the ST data and add more convincing biological analysis.

R. To address reviewer's concern, we now provide to the GEO numbers of ST data (GSE229752; GSE229877) and new TriuSight genomic data (PRJNA1013719). Moreover, we have used orthogonal approaches to validate the ST analysis (multiplex IF, Figure 8) as well as performed an integrative analysis of genotype to phenotype (Supplementary Figure 2).

3. Furthermore, the authors should provide more discussion on the potential clinical implications of their findings. Specifically, they should address how these markers could be used in practice to improve patient outcomes and further validation studies are needed.

R. Based on reviewer's suggestion, we have revised the discussion with the potential clinical implications of our findings.

To date, IPMN management approach must strike a balance between over-utilization of surgery and timely recognition and treatment of patients with high-risk lesions.

Thus, despite the numerous efforts we still lack biomarkers that would reliably allow distinguishing between IPMN that are destined to progress vs those that remain indolent diseases. The validation of the putative biomarkers can only be done within prospective clinical trials. The identification of such biomarkers and their clinical validation were beyond the scope of our manuscript, which has been designed to provide an initial overview of the gene regulatory network and gene expression programs underlying the different IPMN histotypes. We anticipate that our work will spur further investigation on the functional and clinical meaning of those expression programs, e.g. whether they could assist pathologist in determining the fate of IPMNs that have not yet acquired specific features like borderline IPMNs.

4. Although there are new findings in the work, additional experimental validation is necessary to support the findings.

R. See also response to comment to reviewers 1 and 3. As suggested, we validated the ST data by Opal Multiplex Immunofluorescence on IPMN tissue sections (Figure 8). As reported here for the first time, HOXB3 is confirmed to be a marker of LGD IPMN, while the expression of NKX6-2 seems to specifically drive dysplasia increasement of gastric-type IPMN, as described in the meantime of this revision by Sans and colleagues (Cancer Discovery 2023). The protein expression validation by Opal Multiplex Immunofluorescence on IPMN tissue sections confirmed the ST data, thus, although we recognize that further validation study are needed, such as a large randomized clinical trial, we propose that the combined use of 3 binary markers could predict the IPMN type also in the early stages of IPMN

progression. Below is a simple diagram capable of identifying the IPMN types (Supplementary Table S2).

	LGD	BR	HGD GASTRIC	HGD INTESTINAL	HGD PANCREATOBILIARY
HOXB3	1	1	0	0	0
SPDEF	0	1	0	1	0
NKX6-2	0	1	1	0	0

EXPRESSION	
1	YES
0	NO

EXPRESSION LEVELS	
1	LOW
0	HIGH

?

5. Except for very rough spatial trajectories, relevant biological validations have been performed using single-cell or bulk analysis processes, and no cases have required SRT data to complete validation. In fact, from the perspective of the marker genes discovery, single cell or bulk data can also complete the work in this manuscript, so why use SRT data.

R. There are several reasons why SRT data have been used for this study. The main reason is that IPMNs are rarely surgically removed especially in the case of low-grade lesions. Most of these lesions have small dimensions (< 5cm) and have to be carefully evaluated by pathologists to assess histology, invasion, and the possible concurrence of PDAC; so, it is uncommon to avail of fresh or frozen material to perform scRNA-seq. Additionally, FFPE is preferred material to link morphology/histology to molecular phenotypes as histological assessment from fresh-frozen tissue is not necessarily optimal. As we are dealing with different histological types, we deemed the choice of ST as the best experimental approach fitting for the purpose.

At the same time, ST enabled us to easily recover information on all cell types within the lesion while dissociation protocol for sc-RNA-seq might have lead to down-representation of certain cell types as frequently observed for PDAC.

In this study we included the high-grade dysplasia IPMN lesions (associated with, and from which could arise, pancreatic cancer), and the low-grade dysplasia IPMNs that have never led to pancreatic cancer. While for the high-grade dysplasia the tissue collection is active, the low grade-dysplasia are monitored over time and surgically removed only when they already show aspects of clear malignancy thus limiting the material for the study on early steps of transformation. To increase the number of low-grade dysplasia IPMNs, we included in the analysis the low-grade dysplasia IPMNs FFPE samples collected more than 10 years ago (this also allows us to speculate on the actual malignant potential of those lesions and to exclude a malignant development due to the patient's genetic background). For this reason, we opted for an analysis (almost a single cell) on paraffin-embedded tissues.

Regarding the bulk RNA-seq, we think that this technique do not have enough resolution to answer to our specific questions.

We know by experience and literature that tumors with low cellularity, such as pancreatic tumors, or for small and heterogeneous lesions such as IPMNs, bulk analysis could be misleading. An example is given precisely by the molecular classifications of pancreatic tumors (Collison et al, 2011; Moffitt et al 2015, Bailey et al 2016, Puleo et al 2018) which have occurred over the years leading to an increasingly precise classification of pancreatic tumors. In my opinion, these classifications that integrate and refine regularly over time, are also an example of technological progress. Cellularity is therefore a critical aspect of current classification of these tumors (for example ADEX and Immungenic subtypes of the Bailey classification suffer from the presence of non-tumor cellular components). To date, the most precise classification is the one obtained in bulk, but only after microdissection of the tumor cells, which confirms three subtypes linked to intrinsic characteristics of the tumor cells (Pure classic, Immune Classic, Pure Basal) and two classifications (desmoplastic and Stroma) linked to the non-tumor cellular component (puleo et al 2018). For this reason, we decided to use a patient cohort for the identification, in which there is an almost single cell sequencing, and a patient cohort for the validation in which the ROIs are drawn arbitrarily but only on the IPMN's own tissue.

6. Marker genes discovery requires comparison of SRT expression from different grade lesions, which involves comparability of expression data due to the batch effect of SRT data. The author should explain in detail how to eliminate the batch of SRT data to make gene expression comparable. Note that, Harmony method adopted in this paper can only compress SRT data from different sources for cluster analysis, which cannot guarantee the comparability of each gene expression data. Therefore, it is not convincible by the way of identifying marker genes in this paper.

R. Thank you very much for your thorough review. The batch effect is indeed a serious problem that should be addressed carefully in this type of studies, and we understand that is the main concern regarding our study. As suggested, we improved the manuscript adding the missing information in a new paragraph "supplementary material 1". Below is provided a detailed description of the ST analyses.

Visium Analysis 1: Seurat (R)

For most of the analyses, and in particular the discovery of IPMN markers we used two R packages: *i)* ST.Utility and *ii)* Seurat. We uploaded Visium data using ST.Utility wrapper function and after we proceeded with a Seurat analysis. We tried to correct the batch effect with the `sctransform` function regressing out the effect of the different capture areas. However as showed in images below, these function alone was not enough to remove batch effect

from the embedding. Another common effect variable such as the percentage of mitochondrial genes was not taken in consideration as we used the probe based Visium for FFPE that do not detect those genes. We acknowledge that we used few details about the first step of correction in the methods description and we corrected accordingly.

We proceeded to remove batch effect using the SCT assay obtained from `sctransform` to perform Harmony integration using the `Seurat RunHarmony()` function. Thanks to the correction of both counts and embedding we managed to identify clusters that matched the tissue architecture, see Figure 2. Harmony has become the best-practice algorithm for batch effect correction in Visium data, in fact it is recommended by VISIUM manufacturer (10X Genomics, please see the following resource: <https://www.10xgenomics.com/resources/analysis-guides/correcting-batch-effects-in-visium-data>).

Analysis 2. stLearn

To confirm the clusters that we identified with Seurat and to perform spatial trajectory analysis we used the Python packages `stLearn` (revised Figure 7) that takes most of its core functions from the well-known `Scanpy` module. For these analyses we corrected the batch effect using the `Scanpy regress_out` (see figure below) that was inspired and work very similar to `scTransform`; trying to conduct two parallel but somehow comparable analyses. This function corrected the data matrix that was used for all downstream analysis. After, this batch correction we used again Harmony to correct the embeddings and obtain clusters in a similar way that was performed before (see figure below)

As we also think that batch effect may be a principal flaw in this analyses, we tested also other batch effect correction method for python to check the reproducibility of the IPMN clusters and we obtained comparable results, strengthening in our opinion the quality of the data. See below data obtained with ComBat that use integrated empirical Bayes (EB) framework for batch effect correction, that is the best Scanpy-based algorithm for such purpose (10.1186/s13619-020-00041-9).

Analysis 3. GeoMx

For GeoMx data we performed all the QC, normalization, and batch effect correction following all the steps recommended by Nanostring using the GeoMxTools package. More details about the analysis can be found in the following link ([https://github.com/Nanostring-Biostats/ GeomxTools](https://github.com/Nanostring-Biostats/GeomxTools)).

Regarding the doubt about the markers identification, it should be stated that count-matrix batch corrections are not much useful for DE analysis, or even suggested. In fact, it is recommended by Satija lab (the developers of Seurat) to not run Findmarkers function on sctransfrom assay data <https://github.com/satijalab/seurat/issues/4081>.

For our analysis we used DESeq method for markers identification that is implemented Seurat FindMarkers() function that do not use a corrected matrix, however we did not specified in the methods that we used the capture area as the latent variable in the Findmarkers function to account for the batch effect. We corrected the methods accordingly, and we apologize again for the lack of info.

7. GSVA analysis is rough, and relevant results need to be further validated in space, and further explained in terms of biological processes or cell type functions. In addition, it is necessary to discuss whether these finding can be observed in most of the sample data. The marker selected in this paper is mainly TF, and the author should provide discussion on relevant TF regulation network or mechanism.

R. Thank you. We apologize and revised the analysis and the lack of details. We analyzed the entire Molecular Signature Database (Msigdb) interrogating both biological processes and cellular-type signatures. As suggested, we also added the ssgsea calculated with the escape package for each IPMN spot of all the identified gene signatures of interest (Figure 4 and 5 and Supplementary Figure 6 and 9).

Moreover, we assessed the activity of the main transcription factors by SCENIC analysis (Revised Figure 3).

8. Spatial trajectory inference can also be affected by data batch.

In addition, the analysis of spatial trajectory inference is too rough, and relevant details need to be further provided. For example, pseudo-time and their display in situ, the related genes on the pseudo-time heat map display and so on.

R. We acknowledge the difficulties of data analysis due to batch data effect. In the revised Supplementary Methods section, we included a detailed description of methods and how the batch effect was corrected.

In the manuscript we included all the visualizations allowed by stLearn package. We used stLearn pseudotime module specifically suited for spatial data and in particular developed to infer spatial trajectory of cancer cell populations. As requested, we also included a pseudotime heatmap to show the transcription factors expression alongside pseudotime.

9. The authors are unable to provide experimental verification of marker genes, whether to provide predictive analysis of these marker genes in clinical diagnosis. For example, the classification of validation ST data or other types of data.

R. As requested also from other reviewers, we tested and validated data by Multiplex Immunofluorescence on IPMN Tissue Sections on an independent cohort of archival IPMN samples and on normal pancreatic duct. A new picture, confirming the ST data, was added in a panel of figure 8.

Minor issues:

1. Does TMA refer to sections made up of tissue samples from different patients? Whether batch effect exists in one TMA and whether batch effect exists in different TMA?

R. The TMAs are composed by samples from different patients. Please refer to the batch effect correction (please check response to point 2)

2. In Fig 3B, what are the IPMN Signature?

We added details about IPMN signature in the results section and in figure 3..

3. In Fig 7E, what are the clade_69 and clade_11?

R. stLearn use sub-clustering based on both spatial information and transcriptome to infer trajectory. For these sub-clusters the package use the term clade. So clade_69 and clade_11 represent two spatially localized sub-clusters of Borderline IPMN cluster.

4. The legend in the text is too brief, the author should enrich the related legend

R. We edited most of the figure legends to make them more understandable. We apologize for the lack of clarity.

5. There are some formatting or spelling mistakes in the manuscript, which should be checked carefully.

R. We have reviewed the manuscript for grammar, spelling, and formatting errors. Thank you for your attention.

Reviewer #3 (Remarks to the Author): Expert in spatial transcriptomics, cancer genomics, and tumour microenvironment

In this manuscript, to identify novel biomarkers that could better stratify IPMN risk, Agostini et al. performed Visium spatial transcriptomics analysis on 18 core biopsies (4 TMAs) from 14 patients in the discovery cohort. They identified differentially expressed genes and pathways in LGD and HGD IPMN and further carried out GeoMx spatial transcriptomics analysis of 57 IMPN samples from an independent cohort to validate these findings. While this study may present a valuable resource for the community, this study is overall descriptive and the analysis of Visium data lacks depth, the significant analytical and conceptual limitations diminish its value and my enthusiasm for it.

Major comments:

1. In the current version of the Methods section, there's no detailed description of how the batch effects were assessed, how significant they were, and the method(s) applied for batch correction. There's also no evaluation of the performance of their batch correction method. It is known that Harmony is a method that normalizes embedding, but not data matrices.

R. We performed both batch effect corrections, please see supplementary material and a detailed explanation above (see response to reviewer 2 point 6). We improved the manuscript adding the missing information in the paragraph "supplementary material 1".

2A. How did the unbiased approaches for the ST analysis correlate with the annotation of tissue regions by pathologists?

How were correlations with histological features assessed (quantified)?

R. The quality of the clustering was confirmed by both histological evaluation and markers assessment. We added a new supplementary figure 5 to better show cluster correlation with pathological annotation.

We carefully revised the clustering with the help of expert pathologists that confirmed the matching with histological features. They also suggested to assess the expression of the markers that they used in the routine to check the quality of the clusters (Supplementary Figure 3). These markers are general markers of IPMN and do not strongly associate with morphology or dysplasia grade with the only exception of MUC2 that is prevalently expressed in Intestinal IPMN. However, the expression pattern of ST makers matched the positivity commonly found in clinical routine diagnostic.

2B. By eyeballing Fig. 2B and Fig. 1a, it appears TMA3 and TMA4 showed a lower correlation with pathology annotation. Fig. 2B is for visualization purposes and had to eyeball the correlation. Visualization alone isn't sufficient; i would suggest the authors show spot-level correlation.

R. As you pinpointed in the TMA3 and 4, pathological annotations do not correlate for all cores because, as revised in the results section, the IPMN detached (precisely two gastric and three intestinal IPMN) during the Visium procedure. IPMN are a thin layer of epithelial cell that easily tend to detach from both tissues and slides specifically when they are not included inside tissue core like in the case of TMA3 and 4.

3.The study identified 23 spatial clusters. On what basis was the resolution (the optimal number) of their clustering analysis determined? How can they ensure that the tissues weren't overclustered or underclustered?

R. Thank you for your comment. We performed a Leiden clustering with a resolution of 0.85 assuring the optimal number of clustering matching the histological features, higher or lower parameters were tested but led to over or underclustering, resulting in clusters not matching the histological features. We followed a workflow similarly described by Zhang et al. 2021 (<https://doi.org/10.1016/j.cell.2021.10.024>)

4. In Fig. 2A, it's challenging to read the colors. It seems like the HGD intestinal IPMN formed a separate cluster, and HGD IPMN intermixed with ductal tissue.

R. We would like to thank the reviewer for the informative critique as we have realized that the resolution of Fig 2A was not sufficient for cluster visualization. Therefore, we now added a supplementary figure (supplementary figure 5) to improve cluster visualization. Reviewer' s interpretation of fig 2a is correct and biologically consistent with the origin of IPMN which arise from ductal cells. However all the IPMN markers that we have found are not expressed in ductal tissue, please see Figure 3 and Figure 8 (new protein-level validation). It is still debated but in literature and in our opinion not only Intestinal but also Pancreatobiliary represent a more advanced and differentiated type of IPMN than Gastric IPMN and therefore remarkably separated from ductal tissue and gastric IPMN, not only by morphology but also at transcriptomic level.

5. Using the main molecular classification of PDAC (i.e., Moffit activated or normal stroma), how does the Moffit activated signature inform the cell population of these clusters? The Moffit activated signature does not seem to be robust in distinguishing HGD and LGD; for multiple cores, the signature was all over the place. Does the Moffit classical correlate with the PDAC classical lineage state? How about the basal state? Visualization isn't sufficient; please show spot-level correlation.

We would like to thank the reviewer for the informative critique. We added a supplementary figure to show the main markers of each signature at spot level

(supplementary figure 4). We used these signatures to show the reliability of our ST data.

Moffitt stromal activated signature indicates the presence of activated (fibronectin and collagen) producing fibroblasts. These fibroblasts are the main tumor microenvironment cell component of PDAC since its early stages and are the responsible of the desmoplastic stroma (10.1158/1078-0432.CCR-18-1955). This is the reason why this signature was shared by multiple cores and found across all stages of IPMN.

Related to that, we have employed the known classification of PDAC molecular subtypes which were originally derived from expression profiles (either microarray and RNA-seq) of bulk tissues and cell lines. As the reviewer might know, there are two consensus molecular subtypes of neoplastic cells which reflect different degree of fidelity to the pancreatic endoderm. The classical subtype (as defined by Collisson and Moffitt) essentially overlap with the pancreatic progenitor subtype defined by Bailey and is driven by the activity of pancreatic endodermal differentiation markers (described in supplementary figure 4). This is often regarded as the pancreatic cancer with better clinical outcomes (classical type or less advanced). Conversely, progression of PDAC is associated with basal-like/squamous molecular subtypes which display loss of endodermal origin mostly contributed by the epigenetic silencing of endodermal differentiation markers. This subtype is enriched in advanced and metastatic diseases. It is now evident that these two subtypes co-exist within the same tissue, which further highlights the usefulness of spatial transcriptomic approaches to potentially disclose heterogeneous phenotypes. As basal-like/squamous cells accumulate in advanced diseases, we were expecting a lower fraction of cells displaying basal-like/squamous cell state.

Based on reviewer suggestion, we expanded our analysis including data on basal markers (please see supplementary figure 4).

6. What are the differences between the Moffit classical, Collisson classical, and Bailey pancreatic progenitor signatures? How many genes are overlapped among these signatures?

Even if the classifications include different genes they all refers to neoplastic cell states with different degree of fidelity to the pancreatic endoderm. To respond to the reviewer question we are including here the list of genes comprising each of the signatures.

A formal analysis with the regard to the ability of each of the classification system to identify cells displaying similar transcriptional states has been conducted within the TCGA (10.1016/j.ccell.2017.07.007).

The authors convincingly demonstrated that, when applying the 3 different classifications to tumor tissues with high neoplastic cellularity, there is essentially overlap between the classical and pancreatic progenitor as well between basal-like and squamous. The other proposed subtypes (ADEX and immune) emerge when there is an elevated contamination by nonmalignant cells. Of the 3 classification systems, the one proposed by Moffitt and colleagues and generated through virtual microdissection of bulk tumor tissues is widely adopted by the scientific community to stratify PDAC cells.

Please find below a table with the genes belonging to the different signatures.

Moffitt Classical	Collison Classical	Bailey Pancreatic Progenitor	Moffitt Activated	Moffitt Basal
AGR2	AGR2	ABHD2	CDH11	ANXA8L1
AGR3	ATP10B	AHCYL2	COL10A1	AREG
ANXA10	CAPN8	ANKS4B	COL11A1	CST6
BTNL8	CEACAM5	ANXA13	COL1A1	CTSV
CDH17	CEACAM6	ATP7B	COL1A2	DHRS9
CEACAM6	ELF3	B3GALT5	COL3A1	FAM83A
CTSE	ERBB3	B3GALT5-AS1	COL5A1	FGFBP1
CYP3A7	FOXQ1	BTNL8	COL5A2	GPR87
FAM3D	FXYD3	C11orf86	COMP	KRT15
KRT20	GPRC5A	CALML4	CTHRC1	KRT17
LGALS4	GPX2	CARD11	FAP	KRT6A
LRN3	LGALS4	CIDEA	FN1	KRT6C
LYZ	MUC13	CLRN3	FNDC1	KRT7
MYO1A	PLS1	CRYL1	GREM1	LEMD1
PLA2G10	S100P	CTAGE3P	INHBA	LY6D
REG4	SDR16C5	CYP2C18	ITGA11	S100A2
SPINK4	ST6GALNAC1	CYP2C19	LUM	SCEL
ST6GALNAC1	TFF1	CYP2C9	MMP11	SERPINB3
TFF1	TFF3	CYP4F12	POSTN	SERPINB4
TFF2	TMEM45B	CYP4F3	SFRP2	SLC2A1
TFF3	TOX3	DOK4	SPARC	SPRR1B
TSPAN8	TSPAN8	EDN3	SULF1	SPRR3
VSIG2		FAM201A	THBS2	TNS4
		FAM3D	VCAN	UCA1
		FMN1	ZNF469	VGLL1
		FUT2		
		GIPR		
		IYD		
		KALRN		
		KIAA1211		
		KRTAP5-AS1		
		KY		
		LINC01597		
		LPCAT4		

		LRRC66		
		MUC17		
		MUCL3		
		MYO7B		
		NMNAT2		
		NPSR1		
		NPSR1-AS1		
		NR1I2		
		PDE11A		
		PDZD3		
		PHGR1		
		PLA2G10		
		RPL7P31		
		SEMA4G		
		SLC22A18		
		SLC25A23		
		SMPD3		
		TLDC2		
		TM6SF2		
		TMEM253		
		TSPAN3		
		ULK3		
		WSCD2		
		ZDHHC8P1		

Moffitt and Collisson share six genes out of 23, while Bailey Pancreatic Progenitor have a different set of genes.

7. On page 8, the authors stated, “Interestingly, even the low-grade IPMN showed high expression of these signatures, highlighting the presence of classical-like signatures, even in the more indolent IPMN.” I question this as, for all three cases with low-grade IPMN, these signatures weren’t only expressed by low-grade IPMN but also by other tissues. Could there be another possibility? For example, some genes of these signatures are not specific and could also be expressed by immune/stromal cells. Have the authors ruled out such a possibility? Additional validation data would be needed to support this statement.

R. As described previously, Stromal activated markers are expressed in fibroblasts that represent the major cellular population of pancreatic cancer since its early stages. Thanks to the reviewers comment we improved the manuscript with supplementary figure 4 to show that classical and pancreatic progenitor markers are exclusively expressed by IPMN cells, while the stroma activated markers are present in the stroma component.

8. In Fig. 3A, what criteria were used to select these genes? How many transcription factors were these six genes selected from, and how were they identified? Why do the authors find them “interesting”? Please provide more details.

R. The gene markers that we selected were among the top 10 genes resulting from Findmarkers() function using DEseq method. The markers that we displayed in figure 3A were chosen according to log2 Fold Change (>2.5) and p.value < 0.05 and the percentage of expression in the spots above 60% in the cluster to ensure the reliability as markers. We added these parameters to text, and we apologize for the lack of details. The transcription factors that we have identified were never reported in IPMN and this is the reason why we decided to investigate deeper those genes.

9. Both SPDEF and NR4A1 were high in Borderline IPMN, but their expression decreased in HGD IPMN samples. How can their negative correlation with survival be explained?

R. We apologize for any misunderstanding arising from the survival analysis and would like to guide you to comment 10 for further clarification. The survival data, originally displayed, are related to the PDAC cohort within the TCGA, not IPMN.

Our intention with this figure was to highlight how the identified markers were also negative prognostic factors in PDAC patients. Unfortunately, we do not have a biological explanation for this phenomenon, and we have chosen to remove these data from the manuscript. Our best hypothesis is that gene expression and programs may be activated differently at varying disease stages.

10. HGD Gastric IPMN markers PSCA, SDC4, and VISIG1 showed significant variation across the histology subtypes and had the highest expression levels in gastric IPMN. Will this limit their prognostic significance? I suggest stratifying the survival analysis in Fig. 3C by histological features. Will these genes still be significant for survival after adjusting for histology?

R. This is a great suggestion but unfortunately the data showed here are from pancreatic cancer patients included in TCGA and no information about the early lesions are included in the database. We apologize for the misunderstanding and removed it from the revised manuscript.

11. How were these signatures in Fig. 3B built, and how do they differ from those in Fig. 2C?

R. In Figure 2C, we detailed the PDAC molecular signatures previously mentioned in points 6 to 8. The signatures presented in Figure 3B were constructed using the same marker genes of Figure 3A.

In Figure 3B, we utilized this illustration to portray these markers as a signature within a spatial context. To assess gene set activity for these selected markers within each IPMN cluster, we employed the "Addmodulescore()" function from Seurat and visualized the results using a spatial feature plot function.

12. In Fig. 4-5 pathway analysis, can any of these pathways be validated?

R. As recommended by the other reviewers, we corroborated the spatial transcriptomics (ST) data by conducting Opal Multiplex Immunofluorescence on tissue sections of IPMN. This validation process has revealed, for the first time, that HOXB3 indeed serves as a marker for LGD IPMN, while the expression of NKX6-2 appears to be associated with epithelial dysplasia grade, particularly in gastric-type IPMNs. (Please refer to reviewer 2, comment 4, for additional information.)

13. The analysis of Visium data lacks depth. The immune and stromal cell compartments of IPMN are largely overlooked. I would suggest running inferCNV and mapping KRAS mutations using Visium data and, if possible, profiling the immune infiltrates and stromal cell activities.

R. The Visium technology for FFPE samples works on a one-probe-per-gene basis, making it impractical for inferring copy number variations (CNVs) or mutations. However, as also suggested by reviewer 1, we conducted CNV and hotspot mutation analysis of cancer-associated genes using targeted Exome sequencing (TSO500, Trusight Oncology) on our IPMN samples.

Regarding the analysis of immune infiltration, our study was specifically designed to focus solely on the gene expression landscape of IPMN cells. This approach was influenced by the limitations of the Visium technology (10x Genomics), which can only provide gene expression information within a 50-micron square area of a tissue section. This area typically encompasses around 2-6 cells, assuming an average cell diameter of 20 micrometers. Additionally, we used GeoMX technology (Nanostring) for validation, which operates on selected areas of interest (ROIs). For our analysis, we utilized CD45 as a marker to exclude immune cells, allowing us to concentrate on the gene expression profiles of the tumor cells themselves. This approach minimizes the influence of immune cell gene expression on our findings.

14. Why were 23 out of 57 ROIs excluded from subsequent analysis? Even if they aren't covering IPMN, they could be close to IPMN and those ROIs. Could they still be used to study the immune and stromal cells of IPMN?

R. The ROIs were excluded because they did not pass the GeoMx QC post-sequencing, not because they were not covering IPMN. It's worth noting that all ROIs were intentionally created to encompass IPMN tissue. As detailed in our methods section, we did not capture CD45+ areas; instead, we focused solely on PanCK+ cells.

15. Regarding the statement: “HOXB3 and ZNF117 were associated with LGD”. In Fig. 6B, HOXB3 was not included. ZNF117 was included but was also highly expressed in HGD. I would suggest adding normal tissue as a control, validating HOXB3, and conducting further experiments to support the “association”.

R. We appreciate your suggestion. We have revised the manuscript and have adjusted the overstatement regarding ZNF117.

Unfortunately, we couldn't include HOXB3 as a probe for GeoMx analysis. Therefore, as recommended previously, we included normal ductal tissues in Multiplex Immunofluorescence on IPMN tissue sections. This validation confirms that HOXB3 indeed serves as a marker for LGD IPMN, while the expression of NKX6-2 appears to be associated with the increase of dysplasia, particularly in gastric-type IPMNs.

16. Functional validation is needed to support statements such as, “This infers that ZNF117 may be a key transcription factor in the early stages of IPMN,” and, “suggesting that NKX6-2 may be a major trigger of gastric-type differentiation in IPMN.” The same goes for the statements about the pathway analysis. There are many overstatements in the manuscript.

R. We have revised the manuscript to ensure that there are no overstatements regarding ZNF117 and NKX6-2. We appreciate your valuable suggestion.

17. In Fig. 7A, how do these 12 clusters correlate with the clusters in Fig. 2? Why were these stromal cell-enriched clusters distinctly clustered? Could this be biologically relevant or influenced by batch effects? Can the authors interpret the results?"

R. We conducted spatial trajectory analysis using only the two TMA samples that included Borderline IPMNs. This analysis relies on spatial information, not solely on the transcriptome. In reanalyzing these two TMAs with stLearn, we took great care to follow steps similar to Seurat (please refer to our response to comment 6 from reviewer 2). Consequently, the clusters obtained using stLearn may differ from those obtained with Seurat due to the exclusion of the other two TMAs (TMA3 and 4). However, it's worth noting that stLearn confirmed the same IPMN clusters identified with Seurat. The minor distinctions observed in the stroma clusters can be attributed, in our humble opinion, to the absence of the other two TMAs. In the image below, you can observe that the four stroma clusters are characterized by sets of cell-type markers that are also found in normal pancreas by single-cell RNA-seq, thereby excluding the possibility of batch effect influence. As illustrated, the stroma clusters consist of a combination of exocrine, endocrine, and stromal cells (Gene Markers A-C). It's not surprising to find exocrine markers

expressed in all four clusters (Stroma1-4), as exocrine cells are the most prevalent cell type in the pancreas.

A, Exocrine markers; B, Mixed cell markers; C, Fibroblast markers; D, Endocrine markers

When we compare the markers identified in the stroma clusters with the markers of normal pancreas (unaffected by IPMN) from the Protein Atlas scRNAseq dataset, we can observe some similarities. As demonstrated below, *CELA3A* is primarily expressed by exocrine cells in the normal pancreas (as indicated by the Protein Atlas scRNAseq dataset for normal pancreas).

[Redacted Original image taken from:
<https://www.proteinatlas.org/ENSG00000142789-CELA3A/single+cell+type/pancreas>]

However, it's worth noting that exocrine cells in the Protein Atlas dataset are split into different clusters. Interestingly, in our identified Stroma 4, *AMY2B* appears to be expressed only in a subset of exocrine cells (as seen in cluster 9 of the Protein Atlas Dataset). This observation suggests that the varying expression of exocrine markers found in our dataset may be partially attributed to the inherent transcriptomic diversity already present in pancreatic normal tissue.

[Redacted Original image taken from:
<https://www.proteinatlas.org/ENSG00000240038-AMY2B/single+cell+type/pancreas>]

Furthermore, the other stroma clusters (1, 2, 3) express markers that align with specific cellular components. For instance, *PGC* marks stroma cluster 1. According to the Protein Atlas database, *PGC* is expressed by an undefined cellular population that is nonetheless present in normal pancreatic stroma.

[Redacted Original image taken from:
<https://www.proteinatlas.org/ENSG00000096088-PGC/single+cell+type/pancreas>]

Stroma cluster 2 exhibited the highest levels of fibroblast markers in comparison to the other clusters, while conversely displaying the lowest expression of exocrine markers. This observation reflects the presence of a fibrotic stroma reaction of our IPMN samples (please see Figure 1, H&E staining) .

[Redacted Original image taken from:
<https://www.proteinatlas.org/ENSG00000139329-LUM/single+cell+type/pancreas>]

Indeed, Stroma 4 displayed high expression of endocrine cell markers that were enriched in that area.

[Redacted Original image taken from:
<https://www.proteinatlas.org/ENSG00000254647-INS/single+cell+type/pancreas>]

ease, take in consideration that this study aimed to identify the intrinsic cell pathways characterizing IPMN progression. Consequently, the selected IPMN samples exhibit a heterogeneous tissue environment, with some containing proper stroma, others consisting mostly of exocrine tissue, and the majority featuring adipose or dense low-cellularity extracellular matrix (ECM), as depicted in Figure 1. In this context, resolving the stroma population may be difficult without scRNA-seq deconvolution, even using frozen tissues; let alone in FFPE samples. Due to these factors, we exercised caution in naming the different stromal clusters and simply numbered them, as there was no clear histological annotation available.

We sincerely hope that we have effectively addressed all of your major concerns and would like to express our gratitude for your insightful and constructive critiques.

Minor comments

1. What was the median number of genes per spot? This information isn't provided in the manuscript or Methods. Also, what percentage of spots passed QC per core?

R . We provided all the missing info also in the methods section.

2. For GeoMx, in addition to panCK and CD45, any additional (the 3rd) morphology marker was used?

R . We used Syto 13 for nuclei staining. We added the missing information in the methods section.

3. "Five of these clusters precisely defined the different grades of IPMN: the low-grade IPMNN (LGD and Borderline), and the high-grade IPMNS (HGD Gastric, HGD Intestinal, and HGD Pancreatobiliary) (Fig. 2B)." Which five clusters were these?

R . The five IPMN clusters are: LGD IPMN, Borderline IPMN, HGD Gastric IPMN, HGD Intestinal, and HGD Pancreatobiliary.

4. Is panCK able to label all IPMN cells? How sensitive and specific is it for GeoMx analysis?

R. We used the Nanostring proprietary markers (Nanostring GeoMx Morphology Kit) so the PanCK antibody was tested and approved by Nanostring for GeoMx. PanCk label IPMN cells of all stages please see the added multiplex immunofluorescence analysis.

5. In Fig. 6A, please add a color key. In Fig. 6B, please add p-values.

R . We added the missing information.

6. The authors said, “Moreover, we again observed the association between SPDEF and NR4A1 expression with Borderline IPMN.” I didn't find the gene “NR4A1” in Fig. 6B.

R . We apologize for the mistake.

REVIEWER COMMENTS

Reviewer #1 (Remarks to the Author):

The authors adequately addressed previous concerns.

Reviewer #2 (Remarks to the Author):

The manuscript was significantly improved. But there are still a few concerns regarding the data analysis and figure presentation, which should be further addressed.

1. In Figure 3A and 6B, the authors present differential expression analysis of marker genes. However, it is important to consider the potential impact of batch effects on marker gene selection and the comparison of differential expression. While Harmony can eliminate batch effects, it does not address the issue of comparability of gene expression across samples.

2. Despite the effort for addressing the issue of batch removal in the discussion (e.g., Figure XX in the rebuttal), the presentation of relevant diagrams is unprofessional, limiting the information that can be obtained from them. For instance, there is a lack of UMAP display before batch removal and insufficient annotation of cluster results in the situ coloring map.

3. The discussion of batch removal focuses on single-cell transcriptomics methods (e.g., Harmony). However, there are specific batch removal methods available for spatial omics, such as GraphST and PRECAST. The authors are suggested to also include discussion of these methods.

4. While the data provided by the authors has the potential to expand the field of research, the manual labeling should also be considered, and the authors are suggested to provide this information as a key aspect.

5. The authors should carefully review the figures in the main text, ensuring that they are both visually appealing and informative, e.g. ensuring that all necessary components, including corresponding bar diagrams, are included (e.g., Figure 7H).

Addressing these concerns will enhance the clarity, accuracy, and professional presentation of the manuscript.

Reviewer #3 (Remarks to the Author):

The authors have done an excellent job addressing most of my major comments, and I have observed improvements in this manuscript. However, some comments have not been fully addressed:

1. Harmony is a method that normalizes the embedding but not the data matrix. Were the expression values adjusted? If not, how will the batch effect influence the downstream analyses, such as the gene signature score calculation and comparisons?

2A. It is difficult to visually interpret the correlation with pathological annotation based on the data presented in the new supplementary figure 5.

2B. Thanks for the clarification. Among the processed samples, how many experienced this issue? I would suggest excluding those samples with detached IPMN, as for those cases, the pathological annotation may not provide meaningful guidance for downstream analysis.

3. I recommend that the authors include these clustering results obtained using various resolution parameters and their correlations with the histological features. This would help readers better understand the optimal number of clusters chosen.

4. No clustering results were found in the new supplementary figure 5. Was it mistakenly switched with supplementary figure 4?

5. Marker gene expression was not found in the new supplementary figure 4.

6. Thanks for providing the list of genes. A related question is: how many of these genes are consistently detected in the spatial data? Also, did any of the markers used show consistent drop-out across samples?

7. The new supplementary figure 4 seems to be mislabeled, as I couldn't find any expression data.

8. It is unclear if the p-values were adjusted for multiple testing. The authors used p-values instead of adjusted p-values to select significant genes. How many of those genes remain significant after adjusting their p-values? How would such a change influence downstream analysis and their conclusions?

Lastly, the authors should clearly label each figure and suppl figure. The merged PDF file and downloaded reviewer zip file do not contain labels for the figures, which is inconvenient for reviewing.

Point-By-Point Rebuttal

We are submitting the revised version of our manuscript entitled “**Identification of novel spatially resolved markers of malignant transformation in Intraductal Papillary Mucinous Neoplasms**” for potential publication in *Nature Communication*.

We express our sincere appreciation for your contributions in enhancing the quality of our work. We have addressed all raised concerns and are confident that these endeavors have elevated the manuscript's overall quality.

Our thanks to reviewers for their invaluable feedback.

Reviewer #2 (Remarks to the Author):

The manuscript was significantly improved. But there are still a few concerns regarding the data analysis and figure presentation, which should be further addressed.

1. In Figure 3A and 6B, the authors present differential expression analysis of marker genes. However, it is important to consider the potential impact of batch effects on marker gene selection and the comparison of differential expression. While Harmony can eliminate batch effects, it does not address the issue of comparability of gene expression across samples.

R: We regret any lack of clarity in our earlier response. In this revised version, we aim to be clear. We've included a figure illustrating the batch effect correction within Seurat, along with details on the data assays employed by the various Seurat functions.

Seurat workflow

Of course batch effect correction is crucial for this type of analyses. At present, numerous algorithms have been developed for spatial data integration, analysis, and clustering. However, for this particular study, we chose to adhere to the primary recommendations outlined by 10X

Genomics, utilizing Seurat for primary analyses and Harmony for data integration. Our decision stemmed from guidance provided in the tutorial (<https://www.10xgenomics.com/resources/analysis-guides/correcting-batch-effects-in-visium-data>). During a discussion with bioinformaticians from 10X Genomics, whom we had the privilege to engage with, they assessed various algorithms and determined Harmony to be the most effective method for Visium data integration.

Indeed, prior to batch effect correction, identifying clusters shared by all Tissue Microarrays (TMAs) was exceedingly challenging, highlighting the necessity of Harmony for dataset integration. Below, you can observe the complexity of identifying shared clusters between TMAs before implementing Harmony.

You can see (in figures below) how Harmony effectively merged the datasets while still maintaining the distinct biological clusters we identified using cell-type scoring functions to represent diverse cellular populations. In conducting this analysis, we initially executed normalization and variance stabilization by regressing out the varied library sizes (nCountRNA) from each dataset, following the guidelines outlined in this resource (https://hbctraining.github.io/scRNA-seq_online/lessons/o6a_integration_harmony.html). Subsequently, we proceeded with Harmony integration, specifically considering the diverse Tissue Microarrays (TMAs) as batches in the process.

As you highlighted, Harmony integrates the embeddings and not the counts, which, as you astutely mentioned, can impact the outcomes. However, within the Seurat ecosystem, the developers discourage the utilization of integrated counts for analyses not directly associated with clustering. In a discussion highlighted here (<https://github.com/satijalab/seurat/discussions/4000>), they express reservations about comparing integrated counts due to inherent dependencies introduced between data points, which contravene the assumptions of statistical tests used for differential expression.

The decision to use or abstain from a corrected count matrix may hinge more on the algorithms and analyses employed rather than a universally prescribed procedure. For our differential expression analysis, we leaned on a function employing a linear model capable of accommodating differences in library size, rather than relying solely on a simple Wilcoxon test (Seurat's default test). This approach aligns with the strong suggestion presented by Harmony developers in their paper [link], advocating for the robustness of the DESeq2 method over others like MAST, LR, and similar functions, as demonstrated in this study (<https://doi.org/10.1038/s41592-019-0619-0>)

For this reason we used the robust DESeq2 method that was shown to be less prone to false discovery than other functions such as MAST, LR, and other (<https://www.nature.com/articles/s41467-021-25960-2>).

DESeq2 method normalizes the raw count matrix by estimating batch-specific size factors and fitting the normalized counts into a negative-binomial model. To enhance precision of the algorithm we took in consideration only the genes that were expressed in at least the 35% (min.pct parameter= 0.3; 3 fold higher than default parameter) to rule out genes that maybe aberrantly overexpressed due to the batch-effect as recommended by Seurat developers (<https://satijalab.org/seurat/reference/findmarkers>).

To sum up, we adhered to common workflows and standard procedures within Seurat analyses to address batch-effect bias. To further validate the reliability of Seurat results, we not only confirmed consistent markers but also corroborated identical pathways and cell-type-specific signatures using GeoMx, a distinct technology employing different analysis methods. Furthermore, in response to your prior request, we successfully validated the expression of HOXB3, SPDEF, and NKX6-2 at the protein level (Figure 8).

We hope this demonstrates the trustworthiness of our data, showcasing that in Seurat, **integrated counts are not required**, and alternative methods were employed to account for batch effects.

2. Despite the effort for addressing the issue of batch removal in the discussion (e.g., Figure XX in the rebuttal), the presentation of relevant diagrams is unprofessional, limiting the information that can be obtained from them. For instance, there is a lack of UMAP display before batch removal and insufficient annotation of cluster results in the situ coloring map.

R: We regret the previous lack of clarity and the discrepancy in UMAP labels provided in our earlier rebuttal. Enclosed is an updated version detailing the batch effect correction carried out using stLearn.

To validate the clusters identified through Seurat and conduct spatial trajectory analysis, we utilized the Python packages stLearn, primarily drawing upon core functions from the well-

established Scanpy module. For these analyses, batch effect correction was performed using Scanpy's `regress_out` function (refer to the accompanying figure), akin to `scTransform`, aiming to run two parallel yet comparable analyses. This correction adjusted the data matrix used for all subsequent analyses. Following this, we conducted integration using Harmony.

The forthcoming visuals will showcase the distinct clusters identified before and after data integration, each depicting clusters identified using the Leiden algorithm at a resolution of 0.85.

The initial UMAP plot highlights the presence of batch effects influencing clustering before data correction.

Pre-batch effect correction

leiden

The clustering was suboptimal and the majority of IPMN clusters were not identified, and intertwined with the stroma (See cluster 1,5, and 6)

The regress_out function alone could not remove all batch-effects from the embeddings.

Here you can see that the regress_out algorithm was not able to significantly improve the quality of the clustering. However we ultimately resolved the problem with Harmony.

Harmony corrected the embeddings and identified clusters that were comparable with the one obtained with Seurat (included in the manuscript).

We share the concern that batch effects might significantly impact this analysis. To explore this further, we tested additional batch effect correction methods in Python to verify the reproducibility of the IPMN clusters. Our findings yielded comparable results, enhancing our confidence in the data quality. Below is the data obtained using ComBat, which employs an integrated empirical Bayes (EB) framework for batch effect correction (source: [10.1186/s13619-020-00041-9](https://doi.org/10.1186/s13619-020-00041-9)).

Here you can see ComBat alone wasn't able to completely eliminate all batch effects from the embeddings, leading to suboptimal clustering.

Here you can see that only the LGD IPMN (cluster 7) stands out distinctly, while the other IPMN clusters are indistinguishable from the stroma.

The impact of Harmony on our data is clearly evident and remarkable.

Post-Combat Harmony batch-effect correction

leiden

Here, you can observe how even after count correction using the Combat algorithm, the same IPMN clusters were identified. These findings serve as additional confirmation of the quality and reliability of our clustering.

3. The discussion of batch removal focuses on single-cell transcriptomics methods (e.g., Harmony). However, there are specific batch removal methods available for spatial omics, such as GraphST and PRECAST. The authors are suggested to also include discussion of these methods.

R: Thank you also for this suggestion. We added in the discussion section the GraphST and PRECAST as methods for batch effect removal and relative references.

4. While the data provided by the authors has the potential to expand the field of research, the manual labeling should also be considered, and the authors are suggested to provide this information as a key aspect.

R: To confirm the results that we obtained using an unbiased approach, we performed manual annotation of the IPMN clusters discarding the spots that were shared between IPMN and stromal cells and occurred in the IPMN subjected to partial detachment (As suggested also by reviewer 3, comment 2B). The figure below shows the clusters that were manually annotated with an inset depicting the spot positions on the tissue.

Supplementary Information Figure 1

Following manual annotation, the spots underwent normalization and scaling using SCT transform. Differential expression analysis (DEA) was executed using the FindMarkers() function, configuring the DESeq2 method with a min.pct=0.3 (threefold higher than the default parameter). This adjustment aimed to filter out outlier genes that might be influenced by batch effects and consequently expressed aberrantly in only a few spots within the clusters, aligning with recommendations from Seurat developers (<https://satijalab.org/seurat/reference/findmarkers>).

DEA between LGD IPMN and Borderline IPMN, as well as between HGD Gastric and Intestinal IPMN, yielded results consistent with the unbiased DEA analysis. This alignment is illustrated

through Volcano plots and Cneplots, showcasing the expression of the primary signatures previously identified (please see Supplementary Information 2, Figure 2-4).

5. *The authors should carefully review the figures in the main text, ensuring that they are both visually appealing and informative, e.g. ensuring that all necessary components, including corresponding bar diagrams, are included (e.g., Figure 7H).*

R: Thank you, we have revised the images throughout the manuscript by adding the figure names to make the review process smoother. We apologize for the error in Figure 7H; we have included the missing information.

Reviewer #3 (Remarks to the Author):

The authors have done an excellent job addressing most of my major comments, . However, some comments have not been fully addressed:

R: Thank you for your valuable comment.

1. Harmony is a method that normalizes the embedding but not the data matrix. Were the expression values adjusted? If not, how will the batch effect influence the downstream analyses, such as the gene signature score calculation and comparisons?

R: We regret any lack of clarity in our previous response. Please refer to our detailed elaboration in response to major comment 1 from reviewer 2, specifically focusing on batch-effect correction. As detailed in that section, **Seurat developers designed their algorithms to function with non-integrated counts**. This principle also applies to the functions we utilized for the signature scores. For further clarification on the data types employed by Seurat functions, kindly refer to the figure below.

Seurat workflow

As you can see the function AddModuleScoreUCell() and SCENIC, inspired by the AUCell algorithm (<https://doi.org/10.1016/j.csbj.2021.06.043>), use a gene signature scoring method based on the Mann-Whitney U statistic. These scores were create to depend only on the **relative gene expression in individual spots/cells and are therefore not affected by dataset composition and batch-effect**. We used AddModuleScoreUCell() function for gene set activity scoring of pancreatic molecular classification (Moffitt, Collison) and IPMN signatures, while SCENIC was used for Transcription Factor Activity.

Additionally, the runAzimuth() function for cell-type scoring utilized the SCT assay, which represents a normalized and scaled assay corrected for library size using the SCTransform() function. Similar to our approach in Differential Expression Analysis, we diligently followed standard workflows and implemented measures to minimize batch effect influence on our signatures.

2A. *It is difficult to visually interpret the correlation with pathological annotation based on the data presented in the new supplementary figure 5.*

R: Unfortunately, due to the type of acquisition planned for these analyses, the resolution of the images is not excellent. Nevertheless, we have improved the visualization of the annotations by including Hematoxylin and Eosin staining corresponding to the tissue without spot, in the new Figure 5. We believe that this image is now much clearer, and the correspondences are well-visible. Thank you for the suggestion.

2B. *Thanks for the clarification. Among the processed samples, how many experienced this issue? I would suggest excluding those samples with detached IPMN, as for those cases, the pathological annotation may not provide meaningful guidance for downstream analysis.*

R: Thank you also for this suggestion. The Reviewer 2 (comment 4) recommended a manual labeling and analysis of the IPMNs to confirm (and improve) the identification of differentially expressed factors among the various annotated IPMN. For this purpose, and in agreement with your suggestion, we manually assigned the spots only covering IPMN tissues (excluding partially detached IPMN tissue). These results, reported in Supplementary Information 2, confirm the unbiased analyses presented in the main figures of the manuscript.

3. I recommend that the authors include these clustering results obtained using various resolution parameters and their correlations with the histological features. This would help readers better understand the optimal number of clusters chosen.

R: Please here you can find the clustering outcome using various resolution parameters of the Findclusters() function and Leiden algorithm.

In our study we opted for a resolution of 0.85. This is the best parameter to prevent the occurrence of sub- or over-clustering, particularly in IPMN clusters (Supplementary information figure 1).

Supplementary Information Figure 1

However, comparable outcomes were achieved even when configuring parameters within a range with a ± 0.15 difference in resolution from 0.85.

To emphasize the analysis, we also employed broader values (0.65 and 1.05) and extreme parameters (0.5 and 1.2).

For instance, the clustering at extreme broader resolution value (0.65), displayed a discrepancy of only two stromal clusters, while all IPMN clusters remained consistent (see Supplementary information figure 2).

Supplementary information figure 2

Similarly, setting the resolution parameter to 1.05, leads to the identification of two additional stromal clusters, while the clusters for the IPMN were confirmed (see Supplementary information figure 3).

Supplementary information figure 3

Discrepancies between IPMN clusters and histological features were observed exclusively with extreme parameters (0.5 and 1.2) (see Supplementary information figure 4).

Supplementary Information figure 4

The use of extreme parameter (0.5) leads to the clustering of gastric and intestinal IPMNs. While the other IPMNs (LGD, Borderline, and Pancreatobiliary) continue to fall into separate clusters, confirming the different histological features of these IPMN (see Supplementary information figure 5).

Supplementary information figure 5

A clear sub-clustering becomes apparent only in gastric HGD IPMNs when using extreme parameters (1.2). At this resolution value, several subclusters are observed within the epithelium of gastric HGD IPMN, while all the others IPMN fall into separate distinct cluster further confirming their histological features. However, the observed sub-clustering was likely due to the extreme parameter, and no statistically significant differentially expressed genes were found between the two groups using the Findmarkers function (DESeq2 method).

We understand however your concern that we hope now have addressed in full. Thank you again for you carefull review.

4. No clustering results were found in the new supplementary figure 5. Was it mistakenly switched with supplementary figure 4?

And

5. Marker gene expression was not found in the new supplementary figure 4.

R: We regret the errors in our previous response. We revised the figures.

6. Thanks for providing the list of genes. A related question is: how many of these genes are consistently detected in the spatial data? Also, did any of the markers used show consistent drop-out across samples?

R: These signatures are specific markers for pancreatic cancer subtypes. Therefore the expression is related to precise cancer subtype or cell population. However, as specified in comment 1, we used AUCell algorithm to impute the signatures in each spot independently (bypassing batch effect alteration).

7. The new supplementary figure 4 seems to be mislabeled, as I couldn't find any expression data.

R: We apologize. We revised all the figures.

8. It is unclear if the p-values were adjusted for multiple testing. The authors used p-values instead of adjusted p-values to select significant genes. How many of those genes remain significant after adjusting their p-values? How would such a change influence downstream analysis and their conclusions?

R: We apologize for this rough error. Only adjusted p-values were used to select significant genes.

Lastly, the authors should clearly label each figure and suppl figure. The merged PDF file and downloaded reviewer zip file do not contain labels for the figures, which is inconvenient for reviewing.

R: We labeled each individual figure. I hope they are clearer now. Thank you.

REVIEWERS' COMMENTS

Reviewer #2 (Remarks to the Author):

EDITORIAL NOTE: The reviewer only submitted confidential comments to the editor. The reviewer has no further comments or requests.

Reviewer #3 (Remarks to the Author):

The authors have addressed my remaining comments and the manuscript is further improved.